



# Effective uncertainty visualization for aftershock forecast maps

Max Schneider[1,2], Michelle McDowell[3,4], Peter Guttorp[1,5], E. Ashley Steel[6], and Nadine Fleischhut[7,8]

[1]Department of Statistics, University of Washington, Seattle, United States of America
[2]German Research Centre for Geosciences, Potsdam, Germany
[3]Harding Center for Risk Literacy, Faculty of Health Sciences Brandenburg, University of Potsdam
[4]Max Planck Institute for Human Development, Berlin, Germany
[5]Norwegian Computing Center, Oslo, Norway
[6]Food and Agriculture Organization of the United Nations, Rome, Italy
[7]Center for Adaptive Rationality, Max Planck Institute for Human Development, Berlin, Germany
[8]Hans-Ertel-Centre for Weather Research, Offenbach, Germany

**Correspondence:** Max Schneider (maxs15@uw.edu)

**Abstract.** Earthquake models can produce aftershock forecasts, which have recently been released to lay audiences following large earthquakes. While visualization literature suggests that displaying forecast uncertainty can improve how forecast maps are used, research on uncertainty visualization is missing from earthquake science. We designed a pre-registered online experiment to test the effectiveness of three visualization techniques for displaying aftershock forecast maps and their uncertainty. These maps showed the forecasted number of aftershocks at each location for a week following a hypothetical mainshock, along with the uncertainty around each location's forecast. Three different uncertainty visualizations were produced: (1) forecast and uncertainty maps adjacent to one another; (2) the forecast map depicted in a color scheme, with the uncertainty shown by the transparency of the color; and (3) two maps that showed the lower and upper bounds of the forecast distribution at each location. Unlike previous experiments, we compared the three uncertainty visualizations using tasks that are systematically designed to address broadly applicable and user-generated communication goals. We compared task responses between participants using uncertainty visualizations and using the forecast map shown without its uncertainty (the current practice). Participants completed two map-reading tasks that targeted several dimensions of the readability of uncertainty visualizations. Participants then performed a comparative judgment task, which demonstrated whether a visualization was successful in reaching two key communication goals: indicating where many aftershocks and no aftershocks are likely (sure bets) and where the forecast is low but the uncertainty is high enough to imply potential risk (surprises). All visualizations performed equally well in the goal of communicating sure bet situations. But the visualization with lower and upper bounds was substantially better than the other designs at communicating surprises. These results have implications for the communication of forecast uncertainty both within and beyond earthquake science.

## 1 Introduction

Clear communication of uncertainty in forecasts of natural hazards can save lives; likewise, when uncertainty is not communicated, consequences can be disastrous. On 6 April 2009, a magnitude 6.3 earthquake struck L'Aquila, Italy and killed 309



people. The week prior, a senior official from Italy's Civil Protection Agency had urged calm, mischaracterizing the potential for damaging aftershocks triggered by recent earthquakes near L'Aquila. He was later sentenced to six years in prison, along with six leading seismologists[1], with the court concluding that they provided "inexact, incomplete and contradictory information" to the public – in particular, failing to account for the uncertainty in their seismic forecast (Imperiale and Vanclay, 2019).

## 1.1 Aftershock forecasts and their uncertainty

When a large earthquake occurs, more seismic activity is likely. Aftershocks are earthquakes triggered by an earlier earthquake (the mainshock) that can put people at additional risk of harm, for example, by destroying buildings already destabilized by the mainshock (Hough and Jones, 1997). Aftershocks can have a magnitude even larger than their mainshock and can also trigger their own sequences. The spatial rate of earthquakes (the number of earthquakes per unit area) during an aftershock sequence thus follows a highly skewed distribution (Saichev and Sornette, 2007), where many more earthquakes may occur if aftershocks trigger their own sequences of additional aftershocks. The scientific study of aftershock sequences has resulted in sophisticated statistical models (e.g., Ogata (1998)) that can probabilistically describe the expected numbers, locations, times and magnitudes of aftershocks following a mainshock. These models can provide a *distribution* for the number of aftershocks at each location.

Forecasts built from these models are highly sought after by diverse user groups, such as emergency managers and the media (Gomberg and Jakobitz, 2013), in order to inform decisions about disaster declarations and crisis information delivery (Becker et al., 2020; McBride et al., 2020). Recently, several national scientific agencies, including New Zealand's GNS Science (Becker et al., 2020) and the United States Geological Survey (Michael et al., 2020), also began releasing aftershock forecasts to the public. In these public communications, the forecast distribution has been represented in tables of either (1) the expected number of earthquakes above some magnitude for fixed time periods (e.g., one day, one week) after the mainshock or (2) the probability of an event above some magnitude occurring within a time period.

While state-of-the-art models can forecast aftershocks with some degree of accuracy (Schorlemmer et al., 2018), they also have substantial uncertainty. Aftershock models are built on datasets of observed earthquakes in a given seismic region; however, aftershock sequences can vary substantially even within a region, contributing to a large spread in the forecasted number of aftershocks following any mainshock. Forecast uncertainty can be communicated by directly giving this spread for the forecasted number of aftershocks. It can also be communicated implicitly by giving the probability that an aftershock above some magnitude may occur, where a very high or low probability can imply lower uncertainty and a middle probability (e.g., 50%) can imply higher uncertainty. Uncertainty has already been communicated for tabular forecasts by providing these probabilities and ranges of the expected number of aftershocks across the entire forecast region (e.g., Becker et al. (2020)).

Aftershock activity varies over space (Zhuang, 2011) and forecast maps are commonly requested by users (Michael et al., 2020). Fig. 1 shows example maps following the L'Aquila earthquake and the magnitude 7.8 earthquake in 2016 in Kaikōura, New Zealand. Aftershock forecast models will be better calibrated in areas with frequent aftershock sequences than where

---

[1]This ruling was later overturned for the seismologists but not for the civil protection official.



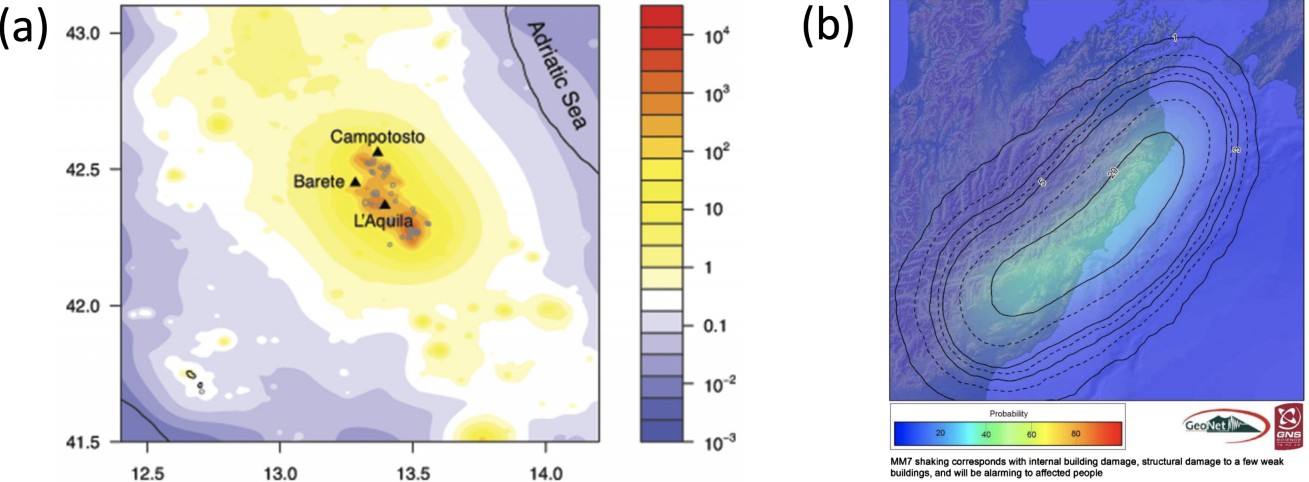

**Figure 1.** Two examples of aftershock forecasts. (a) A map showing the expected daily number of aftershocks above magnitude 2.0 several weeks after the L'Aquila, Italy earthquake of 2006 (reprinted from Murru et al. (2015)). (b) A map showing the forecasted probability of a damaging aftershock (defined as having an Modified Mercalli Intensity score above 7) in the month following the Kaikōura, New Zealand earthquake of 2016, released to the public by New Zealand's GNS Science (reprinted from Becker et al. (2020)).

aftershock activity is sparse, meaning that forecast uncertainty also varies over space. But the uncertainty of the forecast distribution is usually not communicated in aftershock forecast maps; for example, the map in Fig. 1a shows the expected number but not its spread and the probability map in Fig. 1b does not make uncertainty explicit.

As evident in the response to the L'Aquila earthquake, omitting uncertainty from forecasts can affect people's perceptions
and responses to the associated risks. When uncertainty is not displayed, users tend to form their own understanding of where uncertainty is higher or lower, which may not coincide with its actual patterns (Ash et al., 2014; Mulder et al., 2017). Aftershock forecasts that do not communicate uncertainty may therefore result in users misunderstanding the forecast (Fleischhut et al., 2020; Spiegelhalter et al., 2011); for example, in previous studies in the weather domain, users incorrectly expected wind and snow to be lower than forecasted when these had high forecasts, but not when the forecast was low (Joslyn and Savelli, 2010).
Misinterpreting a forecast could become particularly problematic when, due to the skewed distributions of aftershock rates, high uncertainty could mean a greater chance that more aftershocks will occur than forecasted.

While studies across multiple domains have found that displaying uncertainty when communicating forecasts can improve responses related to judgment and decision-making (Kinkeldey et al., 2017; Joslyn and LeClerc, 2012; Nadav-Greenberg and Joslyn, 2009), there is a paucity of literature on uncertainty visualization for spatial aftershock forecasts, or other seismic
communications (Pang, 2008). Thus, despite recommendations to incorporate uncertainty into communications from earth-quake models (Bostrom et al., 2008), there are currently no guidelines on how to do so or on what visualization techniques could help users to understand and incorporate uncertainty to inform their judgments (Doyle et al., 2019). The purpose of the



present study is to develop and evaluate different approaches to visualizing uncertainty in the distribution of spatial aftershock forecasts. These approaches may also be useful for other natural hazards whose forecasts follow a similar skewed distribution.

## 1.2   Visualizing uncertainty for natural hazards

Aftershock forecasts maps can show how many aftershocks are expected for some time period at different locations throughout a region (e.g., Fig. 1a). Uncertainty visualizations have already been designed for similar maps for other geohazards and evaluated using task-based experiments (Kinkeldey et al., 2014). These previous studies can serve as a natural starting point for visualizing uncertainty for aftershock forecast maps. We review the literature that evaluates uncertainty visualizations in geoscientific domains.

One approach to representing uncertainty in geospatial forecasts is by plotting the center of the forecast distribution (e.g., the median or mean) and uncertainty as the spread of the distribution (e.g., the standard deviation or margin of error). The uncertainty is either represented (1) in adjacent maps, where the center and uncertainty are displayed in separate maps or (2) within the same map, using, for example, color, patterns, opacity or symbols to visualize the uncertainty (Pang, 2008). When the forecast and its uncertainty are represented within the same map, designs using color lightness or transparency (i.e., fading the color to a background color like white or gray) have been found to be effective. For instance, Retchless and Brewer (2016) evaluated nine designs for forecast maps of long-term temperature change. They tested an adjacent design against designs that varied color properties (hue, lightness, and saturation) or used textured patterns to display the uncertainty in a single map together with the forecast. Participants had to rank several zones on the map(s), first by their forecasted temperature change and then by their uncertainty. Visualizations where the transparency of the color increased with uncertainty led to more accurate uncertainty rankings than other color-based designs, although the adjacent approach was most accurate.

When forecast uncertainty is represented using the probability of exceeding a threshold value (such as Fig. 1b), studies have also found visualizations using transparency to work well. For instance, Ash et al. (2014) found that a forecast map using red shades of decreasing lightness to depict the forecasted probabilities of a tornado's location were associated with greater willingness to take protective action than forecast maps using rainbow hues or a deterministic map showing only the boundary of the zone of elevated tornado probability (which omitted uncertainty altogether). Similarly, Cheong et al. (2016) found that maps using color hue or lightness to represent the probability, in this case of a bushfire, led to better decisions about whether to evacuate from marked locations on a map (based on realizations of bushfires from a model), compared to a boundary design that omitted uncertainty. A review of dozens of geospatial uncertainty visualization evaluations has similarly concluded that color transparency or lightness can be effective in communicating uncertainty (Kinkeldey et al., 2017).

A third approach to communicate the uncertainty in a forecast distribution is by visualizing the bounds of an interval describing the distribution (e.g., a 95% confidence interval). Although such interval-based maps have recently been proposed, for instance, to communicate public snowfall forecasts by the US National Oceanic and Atmospheric Administration (Waldstreicher and Radell, 2020), there are few studies evaluating the effectiveness of this approach. Nadav-Greenberg et al. (2008) tested uncertainty visualizations that paired a map of median wind speed forecasts with either an adjacent map of the forecast's margin of error, a "worst case" map showing the 90th percentile of the forecast distribution, or boxplots of the forecast distri-





bution at multiple locations. Participants tended to predict higher wind speeds at a given location using the worst case map, relative to the other visualizations. We are not aware of any other studies that have investigated the effects of interval-based uncertainty visualizations for forecast maps.

In the present study, we evaluate three approaches to visualizing uncertainty in aftershock forecast maps. We focus on representing the distribution's spread rather than probability of exceedance because there has been less research on this way of communicating forecast uncertainty (Spiegelhalter et al., 2011) and it is of general interest to other natural hazard forecasts as well. Specifically, we compare a novel interval-based approach where uncertainty can be inferred from the bounds of a 95% confidence interval, a commonly-used approach that uses color transparency to display the uncertainty within the forecast map

and the classic adjacent display. These uncertainty visualizations are compared against a forecast depicted without uncertainty, which is most common in practice and thus the natural baseline (see Fig. 1).

### 1.3   Evaluating the effectiveness of uncertainty visualizations

An effective uncertainty visualization should facilitate not only the reading of the forecast and uncertainty off of a map but also help users to apply this information appropriately. While designing map-reading tasks is more straightforward (e.g., asking

users to read or locate particular areas on a map), previous judgment or decision-making tasks have not been designed to systematically evaluate how users interpret uncertainty given different designs (Hullman et al., 2018; Kinkeldey et al., 2017). Further, without defining tasks in line with specific communication goals, it is not possible to identify what constitutes an effective uncertainty visualization.

    First, tasks should be designed such that the effect of uncertainty visualizations can be disentangled from other features of

the maps or the task. For instance, Viard et al. (2011) asked participants to rank the risk of over-pressure based on an adjacent or a pattern-based uncertainty visualization of estimated pressure of oil reservoirs. Several locations were compared that varied both in mean pressure and uncertainty. While a difference in rankings was found between the visualization conditions, it was impossible to conclude whether either visualization led to more reasonable responses, as the selected locations did not vary systematically and no normative ranking was specified. Other studies have implemented similar ranking tasks using

locations that do not uncover how different designs affect how users understand forecast uncertainty (e.g., Deitrick and Edsall (2006); Scholz and Lu (2014)). Additional issues include tasks that ask for forecast and uncertainty information to be used separately (Retchless and Brewer, 2016), or results that are aggregated across many trials without accounting for the locations' forecast/uncertainty levels (e.g., Cheong et al. (2016)).

    Second, evaluation tasks should be designed to link to specific communication goals relevant across user groups, in order

to inform designs that could serve a wider audience. In Padilla et al. (2017), participants had to decide whether to move an oil rig in the face of an oncoming hurricane, receiving the hurricane track forecast either with a cone of uncertainty or an ensemble of possible hurricane track curves. By systematically moving the oil rig's location across trials, the authors found that decisions were influenced by whether the oil rig was inside the cone of uncertainty or directly on top of an ensemble hurricane track. But the generalizability of these results outside of this highly specific decision task is debatable. Other studies similarly

evaluate uncertainty visualizations with well-designed decision tasks inspired by specialist use cases but it is unclear what their





results indicate for designing public forecast maps (Seipel and Lim, 2017; Correll et al., 2018). In contrast, other experimental literature on uncertainty communication (e.g., Burgeno and Joslyn (2020); Joslyn and LeClerc (2012); Van Der Bles et al. (2019)) works with tasks that are relevant across user groups and designed to reveal how the communication was understood. Visualization scholars have also urged more generalizable tasks in evaluation experiments for uncertainty visualizations (Crisan and Elliott, 2018; Meyer and Dykes, 2019).

Building off this literature, we seek to improve how uncertainty visualizations are evaluated, using an experimental task framed around ubiquitous goals that an uncertainty visualization should ideally achieve. We frame our task around targeted needs users have from aftershock forecast maps.

### 1.4 Communication needs for aftershock forecasts

To understand the informational needs of a common user group for aftershock forecasts, we interviewed five emergency management officials in the United States (see detailed summary in Supplement Text A1). The interviews focused on decision-making for crisis response during natural disasters and on how forecasts could help these decisions, even when this information is uncertain. After synthesizing the interviews, we isolated two commonly mentioned questions that an aftershock forecast should facilitate answering and posited that these communication needs would also be relevant to a general user audience:

1. Where is it likely that aftershocks will or will not take place ("sure bets"; i.e., areas with high/low forecasted aftershock rates and low uncertainty)?

2. Where is a bad surprise possible due to the high uncertainty of the forecast ("surprises"; i.e., areas with high uncertainty, that could yield an aftershock rate higher than forecasted)?

Using these communication needs as a guide, we designed a comparative judgment task to evaluate how different visualization approaches can support these communication needs. Within this task (see Section 2.3), participants judged two locations with systematically varying forecasted aftershock rates and uncertainty levels.

Given locations with *equally low uncertainty* but *different forecasted rates*, users should be able to correctly identify that more aftershocks are expected at the location with the higher forecasted rate. We refer to this as a sure bet trial (addressing communication goal 1). Given locations with *equal forecasted rates* but *different uncertainty*, it is more likely that more aftershocks than forecasted will occur at the location with higher uncertainty, compared to the location with lower uncertainty. As this could lead to a bad surprise, we refer to this as a surprise trial (addressing communication goal 2).

For surprise trials, the forecast level determines whether a response exists that can be considered "correct". When both locations have *low forecasted rates*, users should identify the location with low uncertainty (a sure bet to have few aftershocks) as having less potential for aftershocks compared to the location with high uncertainty. This is because high uncertainty means that in the long run, much higher aftershock rates are possible than forecasted, due to the skewed distribution of aftershock rates. When both locations have *higher forecasted aftershock rates*, it is not possible to define a correct answer for where users should expect fewer or more aftershocks. In this case, comparing locations with lower uncertainty (where high rates are more certain) to locations with higher uncertainty (where even higher rates are possible) may be subjective to each user, for instance,



based on their risk preferences. It is thus even more important to understand how different uncertainty visualizations affect

user judgments within this situation and which visualizations lead users to recognize that forecasts with high uncertainty could result in more aftershocks than forecasts with low uncertainty.

Using the comparative judgment task, we methodically investigate how effectively different uncertainty visualizations fulfill both communication needs. Unlike previous tasks in the uncertainty visualization literature, our task allows to infer which uncertainty visualizations produce responses about uncertainty that are consistent with the forecast distribution and for com-

210 munication needs relevant across user groups.

## 1.5 Research aims and contributions

In the present experiment, we test three uncertainty visualizations for aftershock forecast maps to evaluate which visualization approach best serves the above communication goals. Specifically, we seek to answer the following research questions:

1. How well can lay people read off the forecasted aftershock rate and its level of uncertainty from the different visualiza-
215 tions?

2. How do the visualizations affect people's judgments about where to expect more aftershocks?

   (a) How accurate are people's expectations when comparing locations with varying forecasted rates but low uncertainty, and how does accuracy differ by visualization?

   (b) Where do people expect more aftershocks when comparing locations with the same forecasted rates but different
uncertainty, and how does this differ by visualization?

We evaluate the aftershock forecast visualizations in an online experiment using a broad sample of participants from the United States. Our experiment allows to infer how both classical and novel uncertainty visualizations affect potential lay users' perceptions of aftershock forecasts. Moreover, our work adds to the literature on visualizing uncertainty for natural hazards by using a systematic judgment task based on user needs which may be applicable across hazards.

## 225 2 Method

### 2.1 Participants

Participants were recruited via Amazon Mechanical Turk (MTurk) to complete an online study about map-reading and judgments about future aftershocks using different forecast visualizations. We restricted recruitment to the western U.S. states of California, Oregon and Washington, as these states are seismically active and participants would likely have some earthquake
awareness. While the MTurk population we sample does not match these states' populations, comparisons between MTurk-based and probability-based samples of the U.S. population often yield similar results (Zack et al., 2019), and MTurk has been used to recruit participants in previous uncertainty visualization evaluations (Retchless and Brewer, 2016; Correll et al., 2018). MTurk workers were eligible to participate if they had an approval rating of ≥99% and were using a computer screen ≥ 13





inches in diagonal. We also required participants to answer four multiple choice control questions about the study, based on initial instructions. Participants were given two attempts to answer the questions.

Of the 1392 participants who consented to participate in the experiment, 941 passed the control questions, 908 completed one or multiple tasks and 893 completed the full experiment. Seven participants self-reported to have an age less than 18 and were excluded from analysis, and two participants were excluded because they took too many attempts at control questions, leaving a final sample of 884 participants (46.7% female, median age of 32 years (range: 18-77 years)), binned evenly by condition. The sample size per condition was pre-specified by a power analysis, using effect sizes found in a pilot study.

The experiment was incentivized and participants were informed of this before consenting to the experiment. To calculate incentive bonuses, we randomly drew two map-reading trials and three comparative judgment trials per participant (see Section 2.3). We gave a bonus USD 0.10 for each correct map-reading response and for each judgment response that matched the outcome of a hypothetical week of aftershock activity simulated from the presented forecast map. Including the baseline payout (USD 1.80), participants could earn over U.S. minimum wage for just two correct trials (e.g., the map-reading trials), matching community standards (Paolacci et al., 2010).

We pre-registered the experiment and analysis plan in the Open Science Framework (https://osf.io/2svqk). The study was approved by the Max Planck Institute for Human Development ethics committee.

## 2.2 Materials

### 2.2.1 Creating forecast maps from earthquake model output

Each uncertainty visualization (UV) showed a weekly forecast for the number of aftershocks following a major earthquake together with its estimated model uncertainty. These hypothetical forecasts were created from the output of a spatially-explicit seismicity model (Schneider and Guttorp, 2018), for a seismic zone in the United States. We cropped an area of roughly 2000 km$^2$ from this model output (estimated total seismicity rates) to represent a forecast for the number of aftershocks above magnitude 2.5. Though probabilistic aftershock forecasts would typically be computed from many simulations from a forecast model, our model output was based on most likely parameter values but still maintained characteristic spatial features of aftershock forecasts. In particular, it showed a rapid and isotropic spatial decay from higher to lower forecasted rates, fading into a low (but non-zero) background rate (see e.g., Marzocchi et al. (2014); Zhuang (2011) and Fig. 1). We scaled the model output up to achieve aftershock rates similar to recent major sequences (e.g., 2018 Anchorage, AK and 2019 Ridgecrest, CA, which had 500-1000 aftershocks above magnitude 2.5 in similarly sized zones around the mainshock).

We cropped two distinct ~2000 km$^2$ areas of the map to represent forecasts with different spatial patterns (hereafter referred to as "forecast regions"). Maps were labelled with "Longitude" and "Latitude" along the axes, without tick marks. We randomized experimental trials between these forecast regions to avoid memory effects and allow for more trials, due to expected variability in task response within participants. As the focus of the study was on how to visualize a forecast and its uncertainty, geographical features (e.g., topography, cities, roads) were omitted in forecasts maps to avoid these potentially confounding task responses (Nadav-Greenberg et al., 2008).



### 2.2.2 Visualization of forecasted rate

To visualize *forecasted rates*, the spatial rate distribution derived from the model was mapped onto a grid of $20 \times 20$ cells (each grid cell thus represented an area of approximately 2.8 km $\times$ 1.8 km). We then binned the numeric rates into categories, as is commonly done for natural hazards forecasts and recommended by the visualization literature (Correll et al., 2018; Thompson et al., 2015). We binned rates into five categories, to keep complexity manageable while maintaining sufficient information to display realistic trends. A sixth (uppermost) category was subsequently added that was solely used in the Bounds UV (see below). The category cutoffs came from quantiles of the scaled rate distribution. These cutoff values thus revealed the skewed and concentrated nature of aftershock rates. We refer to this map as the *most likely forecast* map and labelled it "Forecasted Number of Aftershocks". We interpreted each location's most likely forecasted aftershock rate as the center (i.e., the median) of the forecast distribution of aftershock rates for that location (we hereafter abbreviate "most likely forecast of aftershock rate" simply with "rate"). Rates were visualized using a 6-color palette with colors increasing uniformly from yellow to red and ending with brown (see Supplement Text A2 for additional detail on color selection).

Fig. 2 (left panel only) shows the Rate Only condition, which omits forecast uncertainty. This is the current manner of displaying aftershock forecasts to public audiences, when maps are used (see Fig. 1).

### 2.2.3 Visualization of forecast uncertainty

To visualize *forecast uncertainty*, we created uncertainty maps that followed realistic patterns and would also have design features needed for experimental tasks. Uncertainty maps had the same dimensions as the rate maps. Uncertainty values were generated for each grid cell using random number generators and interpreted as the standard deviation of the forecast distribution. These maps adhered to the smoothly varying patterns expected of forecast uncertainty, with higher uncertainties sometimes, but not always, corresponding to areas of higher rate. These uncertainty values were then binned into 3 categories ("Low", "Medium", "High") to allow for sufficient information to display the above-described patterns. In the present article, we present maps and describe experimental materials using an "uncertainty" framing for consistency with terminology used in the uncertainty visualization literature. However, in the experiment, the uncertainty information was presented under a "certainty" framing (e.g., "the forecast is more certain for some locations than for others"), based on feedback from participants during extensive piloting of study materials.

Three UVs were constructed using the rate and uncertainty maps.

### 2.2.4 Adjacent UV

Uncertainty was presented in a separate map adjacent to the rate map (Fig. 2). To display uncertainty, we used the Hue-Saturation-Lightness color model to select three perceptually uniform shades of gray, with constant hue and uniformly increasing lightness.





### 2.2.5 Transparency UV

The Transparency UV (Fig. 3) was developed from the rate map, by changing the alpha levels of the rate colors based on
their uncertainty level. Alpha is a graphical parameter that alters color lightness by fading to white at minimal alpha. Many
previous UV evaluations for natural hazards have found this aspect of color to be an effective visual metaphor for uncertainty
(Cheong et al., 2016; Retchless and Brewer, 2016). We chose the three alpha levels visually to maximize discriminability of the
transparency levels across all colors. We included these in the legend by adding columns of colors with corresponding alpha
for each uncertainty level.

### 2.2.6 Bounds UV

To create the Bounds UV (Fig. 4), we developed two maps: one for the lower bound (most likely forecast minus $2 \times$ standard
deviation) and one for the upper bound (most likely forecast plus $2 \times$ standard deviation) of a 95% confidence interval for each
location's forecasted aftershock rate. While this assumes the forecast distribution is symmetric rather than skewed, it was the
most reasonable approximation available to us without a complete forecast distribution to draw percentiles over. The lower and
upper bound maps were labelled "Optimistic Forecast: Lowest Number of Aftershocks Reasonably Likely" and "Pessimistic
Forecast: Highest Number of Aftershocks Reasonably Likely", respectively. In this visualization, uncertainty is not conveyed
explicitly but can be inferred from the color differences between the lower and upper bounds maps (i.e., a large color difference
at a given location represents greater uncertainty about its forecast). Participants were informed of this interpretation of large
color differences in a short tutorial preceding the study (see Section 2.3). A location with high uncertainty corresponded to a
color difference of at least 4-5 colors between the lower and upper bound maps, regardless of its rate level in the most likely
forecast. That is, for areas of high uncertainty, the pessimistic forecasted rate was much higher than the optimistic forecasted
rate. The exception was in areas where both the most likely forecast and uncertainty is high (see caption to Fig. 4).


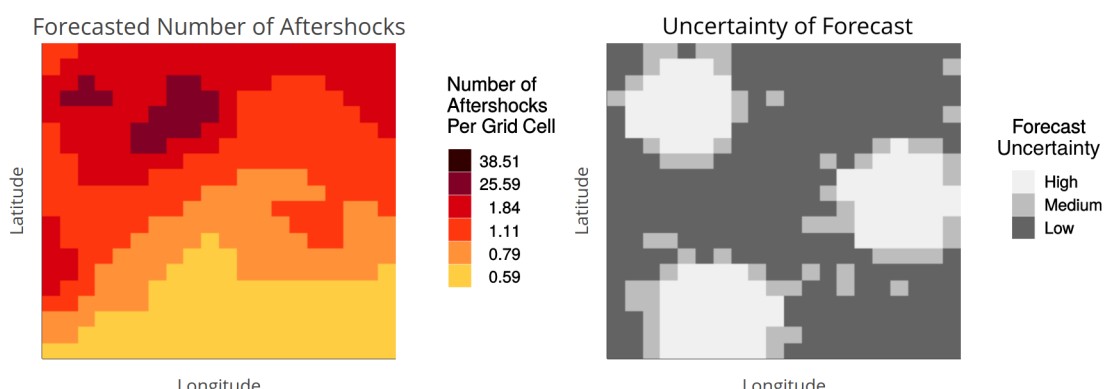

**Figure 2.** Adjacent UV: Most likely forecasted aftershock rate map next to a map of its model uncertainty. Figures 2-4 show UVs for one of the two forecast regions used in the experiment. Since we adopted a "certainty" framing in the experiment, the color palette for the right map (showing certainty rather than uncertainty) proceeded from light to dark for areas of lower to higher certainty, matching the color direction of the rate map. (See experiment screenshots in Figures A1-A3.)

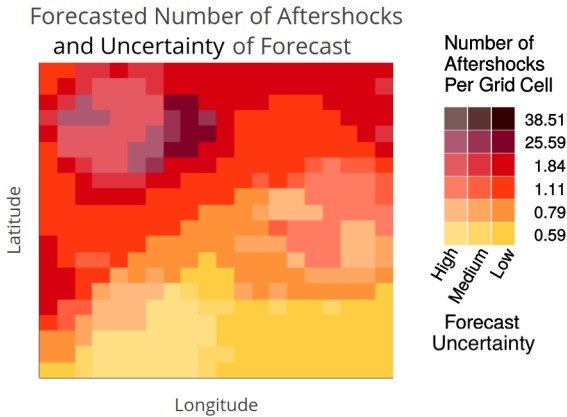

**Figure 3.** Transparency UV: Most likely forecast map in color and uncertainty levels shown by transparency (alpha level) of rate color.


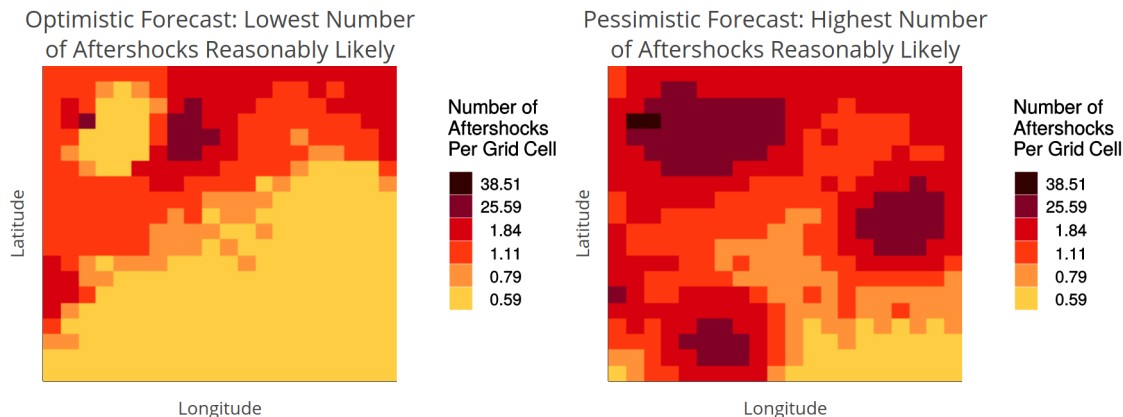

**Figure 4.** Bounds UV: Lower and upper bound maps of a 95% confidence interval around the most likely forecast. The forecast uncertainty at each location is shown through its difference in colors between the maps. Locations with high uncertainty have a 4-5 color difference, regardless of their most likely forecasted rate (e.g., lower left and middle right zones). We designed another scenario for when both the most likely forecasted rate and forecast uncertainty are high: the lower and upper bounds showed red and brown, respectively (e.g., several grid cells in the upper left zone). This scenario represents a case where, due to the high uncertainty, the forecast's lower bound is still very high but its upper bound is extremely high.



## 2.3 Design and procedure

Following consent, participants were randomly assigned to one of the four visualization conditions in a between-participants design (Rate Only: $n=217$; Adjacent: $n=221$; Transparency: $n=221$; and Bounds: $n=225$). Participants were presented with
some basic introductory information about aftershocks, how they are forecast and why forecasts are uncertain. In particular, participants were informed that, in areas where forecast uncertainty is high, many more aftershocks than forecasted could actually occur. This aligns with the skewed nature of the distribution of aftershock rates for spatial grid cells. To ensure they understood this introduction, participants had to correctly answer four multiple choice questions about the information provided. One of these questions related to this interpretation of high uncertainty.

Participants then received a visual tutorial explaining how to read the visualization that they were randomly assigned to. We explained the maps as displaying a forecast of how aftershocks will be distributed across a region in the next week following a major earthquake. We used arrows and highlighting to sequentially introduce the elements of the visualization (rate levels, uncertainty levels, legend), in a standardized way across all conditions.

Following this introduction, participants completed three tasks (see Table 1). Two map-reading tasks evaluated how well
participants could retrieve and integrate the rate and uncertainty levels from the map(s). A comparative judgment task then tested how participants utilized the depicted forecast, together with its uncertainty, to make judgments about future aftershock occurrence.

Within each participant, we randomized forecast region across the two map-reading tasks to counter-balance any potential effects of forecast region on reading the visualizations. For the comparative judgment task, forecast region was randomized
across trials within-participants. To isolate the effect of the rate level on judgment, we repeated trials for low, medium and high rate levels (across both forecast regions), which we set at 0.59, 1.11 and 25.59 aftershocks/grid cell in the most likely forecast map, respectively. To restrict the number of trials, we only focused on low and high uncertainty levels. The rate and uncertainty levels were also randomized across trials within-participants.

**Table 1.** Tasks to evaluate effects of UVs on map-reading (Tasks 1 and 2, for Research Question 1) and judgment (Task 3, for Research Question 2) with aftershock forecasts.

| Task | Instruction | Rate/Uncertainty Levels for Trials |
|---|---|---|
| 1. Read off (variables) | For a particular location on the map, return its rate or uncertainty level. | **Rate**: low (0.59 aftershocks/grid cell), medium (1.11), and high (25.59) rates, with low uncertainty only (3 trials) <br> **Uncertainty**: low, medium, and high rates, with low and high uncertainty (6 trials) |
| 2. Read between (variables) | Mark a location on the map with a particular rate and uncertainty level. | Low, medium, and high rates, with low and high uncertainty (6 trials) |
| 3. Comparative judgment | Given the aftershock rate and uncertainty level, in which marked location will there be more aftershocks in the next week? | Depending on judgment type (see Table 2), locations varied forecast or uncertainty systematically (22 trials) |



### 2.3.1 Read Off task

The first map-reading task required participants to provide the rate level or uncertainty level of a marked location (grid cell). For the Bounds UV, we asked participants to provide the rate level for the marked location on the upper bound map. Each participant provided the rate level for three trials (low, medium, high rate for low-uncertainty locations) and uncertainty levels for six trials (full factorial, see Table 1). Participants in the Rate Only condition were not asked to provide the uncertainty level, as this was not depicted in their map. To increase difficulty, we used locations that bordered other rate or uncertainty levels.

Participants answered using multiple choice response options and responses were scored against the correct answer. Each participant's accuracy (calculated separately for the three rate and six uncertainty level responses) was averaged within visualization condition.

### 2.3.2 Read Between task

The second map-reading task required participants to integrate over both rate and uncertainty variables. In this task, participants 350 were asked to click on a location that matched specific rate and uncertainty combinations. Participants in the Bounds condition were asked for a specific lower bound[2]/uncertainty combination, and participants in the Rate Only condition were asked only for a specific rate level. We asked for 6 locations (full factorial, see Table 1) and switched the forecast region from the one randomly assigned for the Read Off task.

We scored participants' responses on whether the clicked location's rate and uncertainty levels matched what was requested. 355 We again averaged participant accuracy within visualization condition, separately for rate and uncertainty levels.

### 2.3.3 Comparative Judgment task

In the final experimental task, two marked locations were shown and participants were asked to select in which one they would expect more aftershocks to occur in the following week. Specifically, we asked: "Where will there be more aftershocks next week: Location 1 or Location 2? Please make a prediction." We further asked participants to rate their confidence in each judgment, using a six point scale with equidistantly spaced verbal labels: "completely guessing"; "mostly guessing"; "more 360 guessing than sure"; "more sure than guessing"; "mostly sure"; "completely sure" (coded as 1-6, respectively).

We varied the rate and uncertainty levels of locations across trials to evaluate three distinct types of judgment. *Baseline trials* assessed how map features determined judgments. In these trials, participants chose between two low-uncertainty locations with identical rate levels (see Fig. A1 in the Supplement for an experiment screenshot of an example baseline trial). While we do 365 not analyze these to answer our research questions, they provide a control against which to understand the next trials.

*Surprise trials* tested how a change in uncertainty led to a change in judgment. For these trials, we moved one of the baseline trial locations from low to high uncertainty, but kept the same rate (Fig. A2). Thus, each baseline trial yielded two distinct repetitions for each surprise trial (see Table 2), where one of the baseline locations remained constant and the other moved to a high-uncertainty zone. Lastly, *sure bet trials* were used to see how selection changes when rate level changes, but uncertainty

---

[2]Since there were no high and medium lower bound locations that also had high uncertainty, we asked for those two trials on the upper bound map.





is held constant. Participants compared two locations with low uncertainty, where the rate is higher for one and lower for the other (Fig. A3). Thus, sure bet and surprise judgments correspond directly to communication needs 1 and 2, respectively (see Table 2).

**Table 2.** Trials in judgment task by type. Two trial types correspond to UV communication goals. Participants had to judge which of two locations will have more aftershocks in the next week, given the forecast map(s). For each rate level, locations marked with a star/plus sign were identical between the baseline and surprise trials, meaning that only the other location was moved to create a surprise trial.

| | | Location 1 | | Location 2 | |
|---|---|---|---|---|---|
| Trial | Type | Rate | Uncertainty | Rate | Uncertainty |
| 1 | Baseline | Low* | Low* | Low$^+$ | Low$^+$ |
| 2 | Surprise | Low | High | Low$^+$ | Low$^+$ |
| 3 | Surprise | Low* | Low* | Low | High |
| 4 | Baseline | Medium* | Low* | Medium$^+$ | Low$^+$ |
| 5 | Surprise | Medium | High | Medium$^+$ | Low$^+$ |
| 6 | Surprise | Medium* | Low* | Medium | High |
| 7 | Baseline | High* | Low* | High$^+$ | Low$^+$ |
| 8 | Surprise | High | High | High$^+$ | Low$^+$ |
| 9 | Surprise | High* | Low* | High | High |
| 10 | Sure Bet | High | Low | Medium | Low |
| 11 | Sure Bet | Medium | Low | Low | Low |

Each trial type had multiple trials, where we varied the locations' rate levels to assess how this affects the perception of its uncertainty (e.g., surprise trials 2 and 5 in Table 2). This also allowed multiple trials within a rate level (e.g., surprise trials
2 and 3 in Table 2, with low rates). We also repeated each trial in Table 2 across both forecast regions to manage expected within-participant variability in task response. There were thus (3 × 2) baseline trial repetitions, (3 × 2 × 2) surprise trial repetitions and (2 × 2) sure bet trial repetitions. These 22 total trial repetitions were randomized within-participant and the location labels (1 and 2) were also assigned randomly (though recoded in our analysis as described in Table 2).

For baseline and surprise judgments, we attempted to select location pairs with balanced distances to, and symmetry around,
the map center and zones of high uncertainty or high rate. For their trial repetitions, we sought to keep distance between location pairs constant while covering different parts of the map. For sure bet judgments, we again selected locations bordering other rate/uncertainty levels to increase difficulty.

### 2.3.4 Response time and covariates

Since it may be faster to read and use forecast and uncertainty information from particular designs, previous UV evaluations
have measured participant response times (Kinkeldey et al., 2017). To explore UV effects in response time, we recorded





response times for each trial across all tasks to compare the response times of participants in the UV conditions relative to those in Rate Only condition.

Following the judgment task, participants were asked several demographic questions (age, gender, education level, state of residence). We also asked participants how many earthquakes they previously experienced, as past experience has been shown

to affect earthquake forecast perception (Becker et al., 2019).

## 2.4 Statistical analysis plan

We performed confirmatory analysis using confidence intervals to answer our research questions about how UVs affected map-reading accuracy and comparative judgment. In the Read Off task, we aggregated average accuracy in responses about locations' rate levels within each condition. We calculated the 95% simultaneous confidence interval for the mean difference

between each UV condition and Rate Only. We used Bonferonni-corrected standard errors and compared confidence intervals to zero to infer differences between groups. We conducted the same confidence interval analysis for responses about locations' uncertainty levels, comparing across the three UV conditions only and repeated this confirmatory analysis on rate and uncertainty accuracies in the Read Between task. Across all analyses, we use a 5% significance level to determine statistically significant differences between conditions.

We evaluated comparative judgments by calculating the percentage of trials where the participant selects the location with the higher rate or uncertainty (see Table 2), which we then averaged across participants and conditions. Differences between these percentages for UV and Rate Only conditions were computed separately for sure bet and surprise judgments. We again inferred differences between groups using 95% simulataneous confidence intervals. For surprise judgments, we computed differences between UVs and Rate Only specifically for each rate level. This tests whether UV effects on judgments of high

uncertainty locations are consistent when the forecasted rate is low, medium or high.

As an exploratory analysis, we explored whether individual differences affected responses to forecast uncertainty. We built multilevel logistic regression models of participant judgments at the trial level, with the visualization condition and rate level as fixed effects and a random intercept for participant. Model selection using model performance metrics determined other needed explanatory variables (see detailed explanation in Text A5). We specifically investigated how participant covariates and

characteristics of the locations used for judgment trials (distance from map center and from high rate or uncertainty zones) influenced judgment pattern.

Ordinal confidence ratings were analyzed in exploratory fashion across sure bet and surprise judgments and other trial subsets, comparing each UV group to the Rate Only baseline. We also compared how patterns in response times differ by condition and between judgment and map-reading tasks.





# 3 Results

## 3.1 Map-reading tasks

### 3.1.1 Read Off task

Accuracy in reading rate levels was high across all conditions in both map-reading tasks; however, accuracy differed across conditions in reading the uncertainty level (see Table 3 and Fig. 5).

**Table 3.** Percentage of trials answered correctly by condition for both map-reading tasks, separately for rate and uncertainty levels. Participants in the Rate Only condition were not asked to read uncertainty in either task.

| | Read Off | | | | Read Between | | | |
| --- | --- | --- | --- | --- | --- | --- | --- | --- |
| | Rate | | Uncertainty | | Rate | | Uncertainty | |
| | Mean | (SD) | Mean | (SD) | Mean | (SD) | Mean | (SD) |
| Rate Only | 90.8% | (17.8) | - | | 97.7% | (8.0) | - | |
| Adjacent | 92.6% | (19.0) | 94.9% | (15.3) | 94.7% | (12.3) | 95.7% | (13.2) |
| Transparency | 92.8% | (17.3) | 88.1% | (17.9) | 93.8% | (13.1) | 87.4% | (14.6) |
| Bounds | 88.8% | (19.7) | 82.7% | (29.1) | 94.0% | (13.0) | 70.0% | (19.0) |

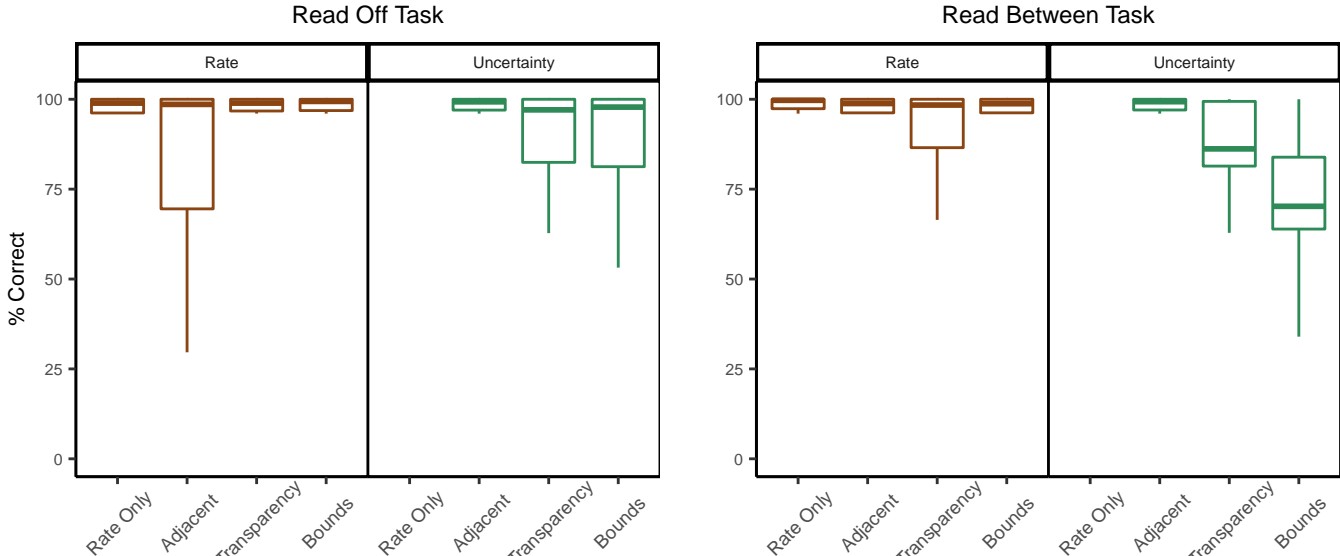

**Figure 5.** Accuracy in the Read Off (left) and Read Between (right) tasks. Boxplots show the percentage of trials answered correctly by participants, grouped by condition. We show results for reading rate and uncertainty separately for both tasks, slightly jittered to reveal trends. Participants in the Rate Only condition were not asked to read uncertainty in either task.





In the Read Off task, participants had a high accuracy in reading off the rate in all conditions, giving correct responses in
between 88.8%-92.8% of trials. Accuracy for participants in the UV conditions was thus statistically indistinguishable from
Rate Only, with differences of less than 2.0 percentage points, on average (see Table 3).

Accuracy in reading off the uncertainty level was lower in the Transparency and Bounds conditions, relative to the Adjacent
condition. Participants gave correct uncertainty responses in 82.7%-94.9% of trials, across all conditions. Compared to the

Adjacent condition, which was most accurate, participants were on average 6.8 percentage points less accurate (95% confidence
interval (CI) = 3.4-10.2) in the Transparency condition, and 12.2 percentage points less accurate (95% CI = 7.5-16.9) in the
Bounds condition. Across all conditions, the majority of inaccurate responses came from misreading the location's uncertainty
level as medium, rather than high or low. This was the case for 57.6% to 80.9% of inaccurate responses within the three UV
conditions.

**3.1.2    Read Between task**

Results were similar for the Read Between task. Participants identified the correct rate across all conditions. Participants in the
Rate Only condition were most accurate for the rate level, and accuracy in UV conditions was no more than 3.9 percentage
points lower, on average.

Just as in the Read Off task, fewer participants identified the correct uncertainty levels in the Transparency and Bounds

conditions than in the Adjacent condition. There was a wider range in accuracy across conditions for the Read Between
task, with participants giving correct uncertainty responses in 69.8%-95.6% of trials. Compared to the Adjacent condition,
participants were 8.3 percentage points less accurate (95% CI = 5.5-11.1) in the Transparency condition, and 25.7 percentage
points less accurate in the Bounds condition (95% CI = 22.4-29.0). The majority of inaccurate responses were again caused by
providing a location with medium uncertainty, across all conditions.

**3.2    Comparative judgment task**

In the comparative judgment task, participants compared two locations that differed systematically in either rate or uncertainty
levels and judged where they would expect more aftershocks. For locations with *different rates* but *low uncertainty* (sure bet
trials), almost all participants in all conditions correctly selected the higher-rate locations (participants selected the higher-
rate location in 92.2%-97.8% of trials). Yet for locations with *differing uncertainty levels* but *identical rates* (surprise trials),

participants in the Bounds condition were more likely than participants in the other two UV conditions to select the higher-
uncertainty location (Table 4 and Fig. 6).

In surprise trials, participants compared two locations with different uncertainty but identical rate levels. In the Rate Only
condition, participants selected the location of higher uncertainty in 52.0% of the trials, on average (see Fig. 6). Note that
uncertainty was not visualized in this condition, so participants saw two locations of equal rate level but may nevertheless infer

differences between them based on other information (Morss et al., 2010). Participants in the Bounds condition selected the
higher-uncertainty location 6.9 percentage points more often (95% CI = 3.2-10.5). In contrast, participants in the Adjacent and




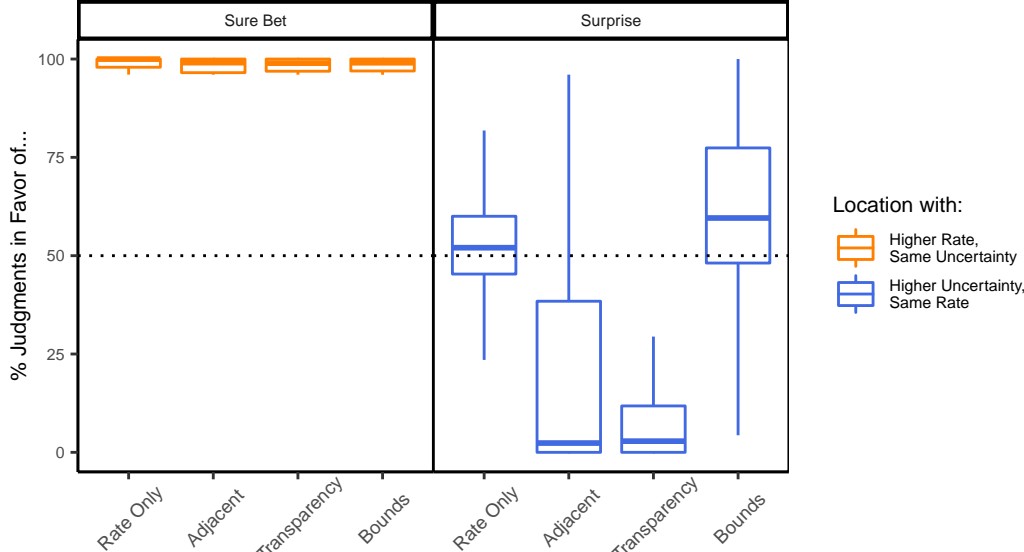

**Figure 6.** Proportion of trials where the location of higher rate (Sure Bet trials, orange) or higher uncertainty (Surprise trials, blue) was selected in the comparative judgment task. Boxplots are over mean proportion across participants by condition, slightly jittered to reveal trends.

**Table 4.** Comparative judgment task: Percentage of trials where participants judged the higher-rate location (sure bet trials) or higher-uncertainty location (surprise trials) to have more aftershocks, per condition.

|  | Sure Bets | | Surprises | |
|---|---|---|---|---|
|  | Mean | (SD) | Mean | (SD) |
| Rate Only | 97.8% | (10.9) | 52.0% | (11.7) |
| Adjacent | 92.2% | (18.1) | 22.8% | (36.3) |
| Transparency | 93.4% | (17.4) | 12.8% | (24.6) |
| Bounds | 95.0% | (14.0) | 58.9% | (22.8) |

Transparency condition selected this location 29.3 (95% CI = 23.8-34.7) and 39.2 (95% CI = 35.3-43.1) percentage points less than Rate Only, respectively.

Moreover, 68.0% of the participants in the Adjacent and 75.7% in the Transparency condition selected the lower-uncertainty location in at least 11 of 12 surprise trials. In contrast, participants' judgments in the Bounds and Rate Only conditions were more variable, with at least 89.0% of these participants selecting both locations multiple times across the 12 trials.

Looking at the surprise trials separately for each of the three rate levels (low, medium, high), the difference between Bounds and the other UV conditions was found for each rate level, but was considerably stronger for trials with low rates and varying uncertainty (see Fig. 7). Participants in the Rate Only condition selected the higher-uncertainty location in 85.9% of the low





rate trials and those in the Bounds condition selected it 1.7 percentage points more often, on average (95% CI = (-3.3)-7.2).
In contrast, participants in the Adjacent and Transparency conditions selected the higher-uncertainty location 59.2 percentage
points (95% CI = 52.6-65.9) and 72.1 percentage points (95% CI = 66.7-77.5) less often, respectively.

The surprise trials with high rates and varying uncertainty contained cases where the higher-uncertainty location had a
possible extreme outcome (high lower bound and extremely high upper bound), which was only visible in the Bounds condition
(see caption of Fig. 4). Participants using Bounds tended to select this higher-uncertainty location for extreme cases (at least
79.0% of participants selected it in these trials) but not for the other high rate trials (7.0% or fewer selected it in these trials),
resulting in their average percentage for all high rate trials to be near 50% (Fig. 7, right).

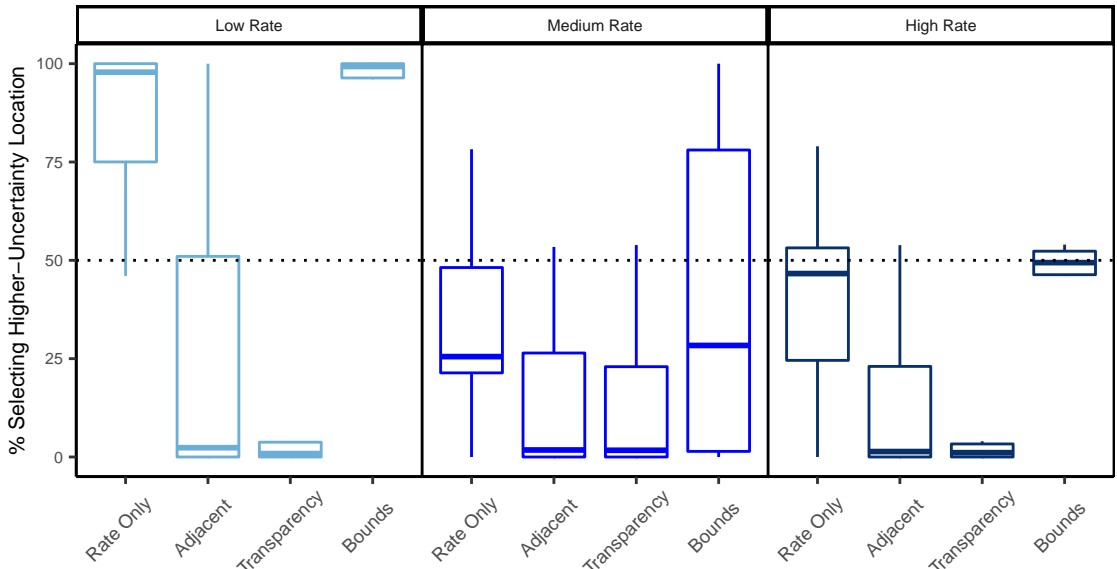

**Figure 7.** Proportion of trials where the higher-uncertainty location was selected between two locations of equal rate level (surprise trials),
grouped by condition and rate level.

### 3.3 Multilevel modeling of judgments

### 3.3.1 Accounting for variability within participants

The effect of visualization on judgments also held after we accounted for variation in judgments on an individual level. We
built a multilevel logistic model for judgments in situations with varying uncertainty and identical rates (surprise trials), with
fixed effects for visualization condition, rate level and their interaction, and a random intercept per participant (see Supplement





Text A5 for details on the model). Because the confirmatory analysis found no evidence of an effect by condition for sure bet trials, we report exploratory analysis on models for judgments in surprise trials only.[3]

While using the Bounds UV slightly raised the probability of selecting the higher-uncertainty location compared to Rate Only, the opposite was true for Transparency and Adjacent. These conditions had significantly negative estimated coefficients (see Table B1 in the Supplement), suggesting, as in Fig. 6, that participants in these conditions were much less likely to select the higher-uncertainty location. This location was more likely to be selected for trials with low rate over medium rate (highly positive estimated coefficient) and high rate, but this effect was dampened by the interaction terms of low rate with Adjacent

and Transparency visualizations (highly negative estimated coefficients). These results match those from the confirmatory analyses in the previous section.

### 3.3.2   Effect of trial characteristics on judgments

Next, we explored whether the spatial patterns of the forecast region and characteristics of the trial locations, such as their distances to important map features, also influenced forecast judgment. These trial-level features did have a measurable effect

on participants' judgments, better explaining the variation in judgments than visualization and rate level alone.

    To assess the importance of these variables, we included the forecast region of each trial, the locations' rate level and their distances to the map center and zones of high uncertainty or rate as fixed effects into the model. We performed a stepwise model comparison between the baseline model and models with these additional fixed effects (see Supplement Texts A3 and A5 for details on the distance measures and the model selection procedure, and Table B2 for model selection results). We again

report results for surprise trials only, as results for models including both trial types were qualitatively similar. The best-fitting model across all four metrics included rate level and visualization, as well as having a location which was closer to the map center or to a high-rate zone (Table 5).

    The UV effects on judgment did not change even when considering the other important variables in this optimal model. The estimated coefficients for visualizations were similar to the baseline model (compare Table B1 to Table 5). Similarly, the

coefficients for rate level and its interaction with UV group were equivalent to the baseline model, reflecting the findings in Fig. 7. The best-fitting model also found that in trials where the lower-uncertainty location was closer to the center or to a high-rate zone, participants tended to select the higher-uncertainty location (highly positive estimated coefficients), though these variables only minorly improved model performance (see Table B2).

    Participants also seemed to use features of the map to make consistent judgments between two locations of the same rate

and uncertainty levels (baseline trials). Fig. B1 in the Supplement shows that even though these locations have the same characteristics, participants tended to select one or the other for some trials, sometimes differently across conditions. Selection in these trials had no consistent pattern based on forecast region or locations' distances to map features.

---

[3]We also built corresponding models using both judgment trials with an interaction term for the trial type, and results were qualitatively similar to those for surprise trials only.





**Table 5.** Most likely estimates and 95% confidence intervals using Wald standard errors for fixed effects in best-fitting multilevel model. The intercept is the logistic of the probability of selecting the higher-uncertainty location for the Rate Only condition and medium-rate trials, with both locations being equidistant from map features (reference level for visualization, rate level and the map features variables). Each fixed effect gives the change in probability of selecting this location from Rate Only to UV conditions, from medium rate to other rate levels, or from equidistant locations to one location being closer to the map feature (all else being held equal).

| Fixed Effect | Estimated Coefficient | 95% CI |
|---|---|---|
| Intercept | -2.56 | [-2.95,    -2.16] |
| *Visualization* | | |
| Adjacent | -1.65 | [-2.18,    -1.11] |
| Transparency | -2.01 | [-2.54,    -1.48] |
| Bounds | 0.68 | [0.22,    1.13] |
| *Rate Level* | | |
| Low | 4.77 | [4.39,    5.15] |
| High | 1.12 | [0.87,    1.38] |
| *Visualization*Rate Level* | | |
| Adjacent*Low | -2.51 | [-2.95,    -2.06] |
| Adjacent*High | -0.92 | [-1.36,    -0.49] |
| Transparency*Low | -3.37 | [-3.82,    -2.91] |
| Transparency*High | -1.61 | [-2.07,    -1.16] |
| Bounds*Low | 0.03 | [-0.39,    0.45] |
| Bounds*High | -0.61 | [-0.94,    -0.29] |
| *Which Location Closer to Center* | | |
| Higher-Uncertainty Location Closer | -0.05 | [-0.26,    0.17] |
| Lower-Uncertainty Location Closer | 1.89 | [1.68,    2.10] |
| *Which Location Closer to High Rate Zone* | | |
| Higher-Uncertainty Location Closer | -0.17 | [-0.43,    0.09] |
| Lower-Uncertainty Location Closer | 1.01 | [0.71,    1.31] |
| *Random effects* | | |
| Intercept $\sigma^2$ | 4.37 | |

### 3.3.3 Individual differences between participants' judgments

To explore whether the participant-level variables we measured could account for differences in surprise judgments, we considered additional fixed effects for the number of earthquakes participants previously experienced, education, age, gender, and state of residence, performing the same model comparison as described above. None of these individual differences improved the model fit across multiple metrics (see model comparison in Table B3 in the Supplement). Thus, differences in visualization and rate level explain participants' judgments better than any of these participant-level variables.





We also found no systematic differences on these variables between participants who consistently selected either location in
11 out of 12 trials and those with more variation in judgments.

### 3.4   Confidence and response time

### 3.4.1   Confidence ratings

We asked participants to rate their confidence after each judgment and we investigated these by visualization condition and
between sure bets and surprise judgments.

We calculated median confidence ratings for each participant within each trial type. Confidence did not differ by condition
for the sure bet judgments, with identical medians and equivalent distributions. For the surprise judgments, participants in the
Rate Only condition (who did not see the difference in uncertainty between the two locations) were generally less confident
than participants in the three UV conditions. The median confidence rating was 2 points lower for Rate Only compared to all
UVs.

We also investigated whether surprise judgments were made more confidently in favor of higher- or lower-uncertainty lo-
cations, as suggested in Hullman et al. (2018). Again, we calculate median confidence ratings per participant for those trial
subsets. Participants using Rate Only and Bounds had similar confidence regardless of their judgment; however, confidence
appeared to differ by judgment for participants using Adjacent and Transparency (see Fig. B2 in the Supplement). When they
selected the higher-uncertainty location, participants in these two conditions had median confidence ratings that were at least
1 point lower than when they selected the lower-uncertainty location. The spread of their confidence ratings was also lower
(shorter boxplots) in those trials where they selected the lower-uncertainty location.

### 3.4.2   Response times

We also investigated trial response times by calculating medians within participant across trials, separately for each of the
three tasks. There was no meaningful difference across the UV conditions in the Comparative Judgment and Read Off tasks,
with differences in median response times to Rate Only of less than 1.5 sec, across all trial types (figure omitted). In the Read
Between task, response times were slightly shorter for participants in the Rate Only condition (who were only asked to identify
a particular rate level) than participants in the UV conditions (who were asked to identify a particular rate and uncertainty
level).

## 4   Discussion

Uncertainty visualization is critical for forecast communication, as studies have shown that communicating uncertainty im-
proves user decision-making and prevents users from building their own representation of where uncertainty is high or low
(Joslyn and LeClerc, 2012; Ash et al., 2014). We compared three visualizations of uncertainty for aftershock forecast maps
against a visualization of the forecast without uncertainty. The experimental tasks were designed to systematically evaluate





how well users could read off the forecasts and how effectively the visualizations served two user-generated communication needs. In particular, aftershock forecast maps should communicate where the certainty is high that aftershocks will or will not occur, as well as where outcomes worse than forecasted are possible, due to high uncertainty.

The results show that all three uncertainty visualizations led to correct judgments about where to expect more aftershocks when both locations had low uncertainty and a difference in forecasted aftershock rate. However, the uncertainty visualizations resulted in significantly different judgments when the locations had forecasts where the uncertainty varied. Although users of the visualization showing the bounds of a forecast interval (Bounds UV) could read off the uncertainty less accurately than the other visualizations, it was the only one where users demonstrated an understanding that forecasts with high uncertainty could have outcomes worse than forecasted.

## 4.1 Effects of visualization

While there was no difference in how well participants could read the rate information from the different visualizations, accuracy in reading the uncertainty information differed depending on the visualization participants saw. Consistent with previous research, the adjacent design was associated with greater accuracy compared to the transparency-based design (Retchless and Brewer, 2016) and the interval-based design (Nadav-Greenberg et al., 2008). These differences held across both map-reading tasks. The majority of inaccurate responses in these tasks, across all visualizations, were in misreading the uncertainty level as being medium rather than high or low. For the Transparency UV, participants may have had difficulty distinguishing between the non-transparent version of one color (low uncertainty) and the more transparent version of the next darker color (one transparency level lighter, i.e., medium uncertainty). For the Bounds UV, since uncertainty was not depicted explicitly in the legend, participants may have had trouble classifying a location into the correct uncertainty category. Consistent with past research, aftershock forecast users were most accurate reading uncertainty information off adjacent displays compared to those that used transparency or represented uncertainty implicitly through a forecast interval.

In the judgment task, there were differences by visualization in how participants judged higher-uncertainty locations compared to lower-uncertainty locations. Most users of the adjacent and transparency visualizations expected the lower-uncertainty location to have more aftershocks than the higher-uncertainty location. This judgment pattern was so consistent that the majority of these participants selected the lower-uncertainty location in at least 11 of the 12 trials. In contrast, users of the Bounds UV were more likely to expect that the higher-uncertainty location would have more aftershocks, relative to the Adjacent and Transparency UVs.

Users of the Bounds visualization might have had this different judgment pattern, compared to the other visualizations, as a result of Bounds explicitly displaying the extremes of the forecast distribution rather than the uncertainty; the uncertainty must be inferred by comparing the lower and upper bound maps. The word uncertainty was also not used in the legend in the Bounds visualization, and we instructed participants that locations with higher uncertainty can only be identified by their large color differences between the two maps. In contrast, the other uncertainty visualizations explicitly depicted the uncertainty and required users to infer where high uncertainty could yield extreme outcomes.



There are three potential explanations for why explicitly depicting the extremes rather than the uncertainty could account for our results. First, consistent with the results of Nadav-Greenberg et al. (2008), the upper bound map may have served as an anchor point that biased users' perceptions towards higher values. Users of the Bounds visualization may have expected more aftershocks in higher-uncertainty locations because they focused on the worst case scenario map (the upper bound) and did not even pay attention to differences in color to infer the uncertainty.

Second, the explicit depiction of uncertainty in the adjacent and transparency designs may have led users to associate high uncertainty with low forecast quality. That is, despite the study instructions explaining that locations with high uncertainty could have worse outcomes due to the skewed distribution of aftershock rates (which participants had to answer correctly before continuing to the study), participants may have nevertheless interpreted lower-uncertainty locations as being more reliable. Furthermore, uncertainty may have been interpreted differently based on whether the visualization presented it with words or numbers. Consistent with previous research (e.g., Retchless and Brewer (2016)), we used an ordinal uncertainty scale with verbal labels ("Low", "Medium", "High") in the adjacent and transparency designs. In contrast, the interval-based design used numeric legends in both lower and upper bounds maps. Previous experiments in non-spatial uncertainty communication have found that verbal labels can lower perceptions of forecast reliability compared to numeric ranges (Van Der Bles et al., 2019), which might explain the difference in judgments between the uncertainty visualizations.

Third, differences in judgment patterns between the visualizations may have resulted from differences in how the uncertainty visualizations used color to signify uncertainty. Higher-uncertainty areas are always marked by darker colors in the upper bound map than lower-uncertainty areas of the same rate level. In contrast, higher uncertainty is marked by lighter colors in both the adjacent and transparency visualizations. These differences in color lightness may have affected how participants interpreted higher uncertainty in the visualizations. Previous studies have reported a dark-is-more bias in how people interpret color scales (Silverman et al., 2016; Schloss et al., 2018). Users of the adjacent and transparency visualizations may have thus perceived the lighter-colored zones of high uncertainty as having lower potential of aftershocks.

## 4.2 Effects of rate level

While users of the Adjacent and Transparency visualizations had the same patterns of selecting the higher-uncertainty location regardless of its most likely aftershock rate (rate level), judgment differed by rate level for the Bounds visualization. When the compared locations had *low rate levels* (most likely aftershock rate = 0.59 aftershocks/grid cell; yellow color), the majority of participants using the Bounds visualization correctly expected more aftershocks at the higher-uncertainty location, which had the much higher upper bound. That is, in these comparisons, the lower bounds of both locations were of equal color (both yellow) but their upper bounds showed a near maximal color difference (from yellow to red), indicating the higher-uncertainty location's potential for much higher rates.

When the compared locations had *medium rate levels* (most likely aftershock rate = 1.11 aftershocks/grid cell; middle orange color), participants' judgments varied between the lower-uncertainty (showing a medium rate in both lower and upper bound maps) and higher-uncertainty (showing low rate in lower bound, high rate in upper bound map) locations. When the compared locations had *high rate levels* (most likely aftershock rate = 25.59 aftershocks/grid cell; second-darkest red color), participants





only selected the higher-uncertainty location when the upper bound had an extremely high rate (38.51 aftershocks/grid cell; brown color, see Fig. 4). In these situations, the lower-uncertainty location always showed a high rate (red color) across both lower and upper bound maps, indicating that a high number of aftershocks is very likely to occur. When the upper bound for the higher-uncertainty location showed an extremely high rate (brown color), participants were likely to select it. However,
when the upper bound of the higher-uncertainty location only showed a high rate (red color) and the lower bound showed a low rate (yellow color), they almost never selected the higher-uncertainty location.

These results indicate that the Bounds visualization led to judgments that recognized the relationship between high uncertainty and the potential for outcomes that could be worse than forecasted. For judgments where the higher-uncertainty location was much higher in the upper bound than the lower-uncertainty location, the majority of users of the Bounds UV selected the
higher-uncertainty location. Users of the other uncertainty visualizations, who did not see the extremes explicitly, made the opposite judgment. This difference between uncertainty visualizations held even when accounting for within-participant variability and other potential participant-level determinants of judgments. Thus, the estimated visualization effects were robust across the sampled population, with respect to the studied covariates.

If highlighting the potential for (even) higher aftershock rates in cases of high forecast uncertainty is critical for a decision at
hand, then our results support displaying forecast uncertainty with maps showing forecast intervals. Where locations have the same low uncertainty, the higher-rate location may be interpreted to have more aftershock potential and where locations have the same rate level, the higher-uncertainty location may be interpreted to have more aftershock potential. In contrast, adjacent and transparency-based displays appear to lead to an opposite response to high uncertainty.

### 4.3   Effects of map features

Previous studies have found that risk perception about a location can be impacted by its distance to risky areas (Ash et al., 2014; Mulder et al., 2017). We also found some evidence indicating that map features of the trial locations influenced judgments using aftershock forecasts. The model selection analysis found that when a lower-uncertainty location was closer to the map's center or to a high rate zone, the higher-uncertainty location was slightly more likely to be selected, irregardless of visualization (see Table 5). Furthermore, participants were not evenly binned when comparing locations of equal rate and uncertainty, with clear
differences by visualization in some trials (see Fig. B1). It is possible that other map-related features that we did not measure could account for these differences in judgments. Future research could explore systematically how judgments are affected by map features or location characteristics.

### 4.4   Judgment confidence and response time

We found only minor differences between visualizations in user confidence in judgments. Not surprisingly, confidence rat-
ings were higher when comparing low-uncertainty locations with different rates, than locations with different uncertainties. In general, participants using the forecast depicted without uncertainty made judgments between locations with different uncertainties with lower confidence than those using UVs. This suggests that omitting uncertainty lowers confidence for judgments between two locations when uncertainty differs but rate does not. We observed higher and less variable confidence ratings





for judgments in favor of the lower-uncertainty location, compared to those for the higher-uncertainty location, but only for

users of the adjacent and transparency visualizations. These designs appear to encourage a consistent interpretation of uncertainty, leading lower-uncertainty locations to be confidently associated with more aftershocks. Response time distributions were equivalent between visualizations, meaning that no evidence was found to suggest visualizing uncertainty increases the time needed for map-reading or making comparative judgments. This result mirrors findings summarized in a review of previous UV evaluations (Kinkeldey et al., 2017).

**4.5    Limitations and future research**

Our study sought to examine the effects of UV designs on specific communication needs for aftershock forecast maps. To do so systematically, we had to fix one variable, either the forecasted aftershock rate or its uncertainty, in the comparative judgment task. In real-world decisions, locations must often be compared where both the rate and uncertainty vary, for example, comparing a location of medium rate/low uncertainty against one of low rate/high uncertainty. Future studies should explore

these comparisons by systematically testing location pairs with meaningful differences, as recommended in prior reviews (Hullman et al., 2018; Kinkeldey et al., 2017).

Geographical features, such as roads and landmarks, were omitted from our maps in order to avoid potential confounding effects on judgments, as in Nadav-Greenberg et al. (2008). However, omitting these features lowers the ecological validity of the study as geographical features are generally included in public forecast communications. Future experiments could add

standard map layers to visualizations and evaluate their effects on interpretations of the forecast made using an uncertainty visualization. We found some evidence that map features influence forecast perception, especially when locations with the same forecasted rate and uncertainty were compared. Controlled experiments should target the effects of map features on task response, both with and without additional map layers.

Finally, our evaluation used a single judgment task and only considered three uncertainty visualizations. More tasks would

elucidate various other effects of candidate uncertainty visualizations, especially tasks that move beyond judgment and towards decision-making. For example, if aiding resource allocation is a communication goal, participants could be asked to allocate search and rescue teams across the map, based on the forecasted rate and its uncertainty. As aftershock and other natural hazard forecasts are increasingly released via public portals with interactive capabilities (Marzocchi et al., 2014), tasks with interactivity can assess how this moderates an uncertainty visualization design's effects. A greater variety of uncertainty

visualizations designs can also be evaluated, particularly those that represent uncertainty using patterns or other overlays easier to separate from the forecast's color than color transparency. Expressing uncertainty in a way that does not use color lightness could investigate whether the dark-is-more bias or something else affects the interpretation of high uncertainty.

**5    Conclusions and practical implications**

Our experiment found that the three approaches for visualizing uncertainty in an aftershock forecast map differed substantially

in their effects on map-reading and judgment responses. These visualizations included one of the most effective designs from





the visualization literature (color transparency) and a newer approach with recent operational use (forecast intervals). Our work suggests several practical implications for the design of aftershock forecast maps for non-technical users. If the accurate reading of uncertainty is the most important aim (e.g., for technical users who may need the aftershock forecast distribution as a direct input into their own models), our results support communicating this explicitly and with a separate and adjacent map. Map-reading accuracy was high across all visualizations, but uncertainty was read most accurately for adjacent designs, compared to transparency- and interval-based designs.

If instead the aim is to communicate that higher-uncertainty locations could lead to more rather than fewer aftershocks, and some inaccuracy in the (implicit) communication of uncertainty is acceptable, then our results support representing uncertainty using forecast intervals. While all visualizations were able to convey that higher-rate locations are sure bets to have more after-shocks when uncertainty is low, only the intervals-based design communicated that high uncertainty locations were potential surprises to have more aftershocks than were forecasted. As this interpretation of high uncertainty is consistent with the skewed distribution of aftershock rates, the intervals-based visualization may improve judgments by non-technical users. For example, emergency managers rely on aftershock forecasts to decide whether to issue a disaster declaration during an aftershock se-quence or not, as in the case of L'Aquila, Italy. The intervals-based design may help such decisions to be more consistent with the skewed distributions of these forecasts. Furthermore, our results have the potential to transfer to visualizations for other natural hazard forecasts that also follow such skewed distributions.



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

**Appendix A: Supplemental descriptions and figures**

**A1    Summary of interviews with emergency managers**

We interviewed five emergency management officials from Washington State, USA, four of whom were directors of either state or county-level emergency management offices (the fifth participant was a volunteer at a county-level emergency management office). The interviews focused on how scientific information is used when responding to natural disasters, and how visual features of previous disaster scientific communications affected how the emergency managers used them for decision-making

needs. We also asked about how uncertainty in scientific communications affects their use for disaster response. We began by posing general questions and then narrowed in on previous experiences with non-earthquake hazards, as most participants had not yet worked in earthquake response. Interviews were recorded and then summarized across participants, finding similarities and differences across the individual responses. We identified several important use cases and communication goals, around which we structured the experimental tasks to evaluate uncertainty visualizations.

**A2    Selection of colors for maps**

The yellow-orange-red (Yl-Or-Rd) palette is often recommended for natural hazards (Doore et al., 1993) and has appropriate color connotations across Western cultures, especially for the lowest rate level (yellow, commonly connoting caution in the natural hazards context) and high rate level (red, commonly connoting danger) (Sherman-Morris et al., 2015; Thompson et al., 2015). We began with the Yl-Or-Rd RColorBrewer palette, known to be colorblind-friendly (Brewer and Harrower, 2009).

We fixed the start and end colors as suggested by this palette: respectively, yellow (made darker to maintain distinguishability for different transparency levels) and a dark red. We then created a five-color palette that was perceptually uniform using the Hue-Saturation-Lightness color model. We uniformly decreased colors' lightness and increased hue, while keeping maximal saturation. The hue and lightness of all colors were then adjusted to be optimally discriminable in both the legends and maps. The sixth color (highest rate level) was made dark brown, a color off the Yl-Or-Rd color palette to further distinguish it from the





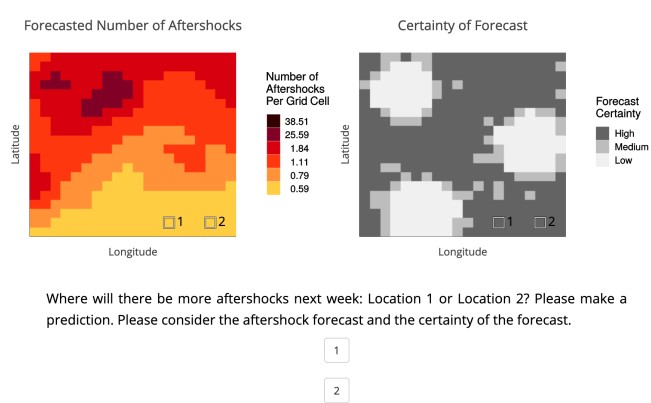

**Figure A1.** Screenshot of baseline trial for Comparative Judgment task with the Adjacent UV.

previous colors, as it was not in any visualization aside from the intervals-based design. This was to avoid the false matching of the darkest color in the map with the darkest color on the legend, as occurred when piloting.

We used a white color with black shadow for the area marker and label in the Read Off and Comparative Judgment tasks to avoid simulatanous contrast affecting the perception of the color within the marker (Krauskopf et al., 1986).

## A3 Developing location distance measures

We investigated whether the characteristics of trial locations in the comparative judgment task influenced how participants judged between them. We focused on primary map features (map center, zones of high-rate and -uncertainty) and whether either location was substantially closer to them.

For each location, we calculated its Manhattan distance (number of grid cells) to the map center and the nearest zone of high rate or uncertainty. We then calculated the difference between these distances for the two locations in each trial. These distance differences are only meaningful when neither location itself has high rate or uncertainty and were only computed in those cases. Furthermore, we are interested in assessing whether locations were meaningfully closer to a given map feature, as we did not expect participants to be influenced by small differences in distances. Thus, we created a new categorical variable indicating which location was at least 3 grid cells (the median distance) closer to each map feature. If both locations were essentially equidistant from that map feature (difference of 2 grid cells or fewer) or if either location was itself in that zone, the trial's value of this variable was "neither", which was set as the reference category. These categorical variables entered our model selection procedure as potential fixed effects (see Text A5).

## A4 Screenshots of conditions from experiment platform





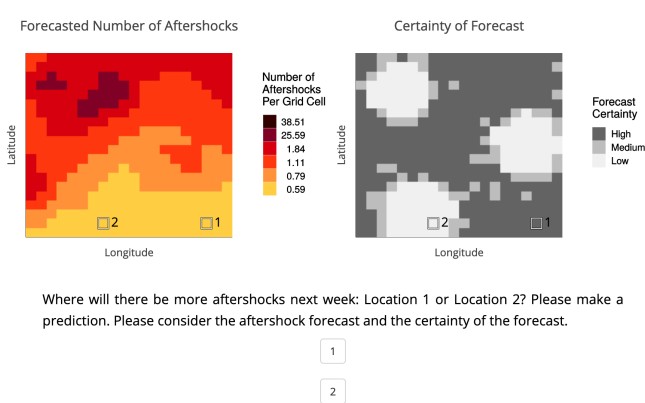

**Figure A2.** Screenshot of surprise trial for Comparative Judgment task with the Adjacent UV.

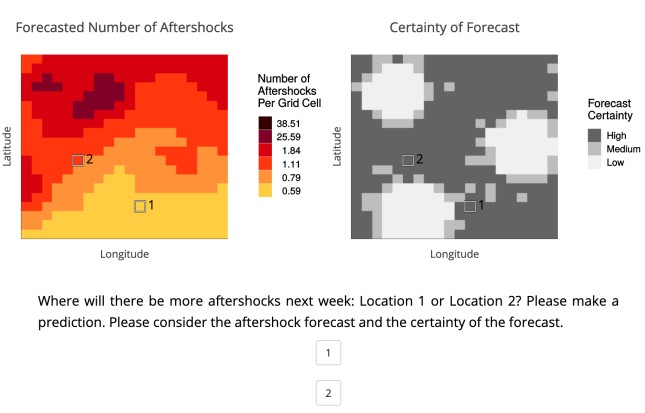

**Figure A3.** Screenshot of sure bet trial for Comparative Judgment task with the Adjacent UV.





## A5  Multilevel model and model selection

We expected participant heterogeneity in comparative judgment, irrespective of visualization condition. We did not experimen-
tally control for the myriad sources of this heterogeneity, nor did we know *a priori* which sources will dominate judgment, so
we omitted participant-level effects from our main confirmatory analysis. We instead built a multilevel model of participant
judgments to explore this heterogeneity by participant. We then performed a model selection analysis to identify drivers of
task response when accounting for individual differences. We investigated which participant- or trial-level variables led to best
model fit (across all participants and trials).

We first built a baseline multilevel regression model on $Y_{ij}$, (participant $i$'s judgment in trial $j$) with a fixed effect for
condition, rate level, and their interaction as well as a random participant-level intercept. We used a treatment contrast for
the visualization condition, with reference category being Rate Only. The baseline multilevel model (built using the glmer R
package, version 1.1-27) is:

$$logit(P(Y_{ij}=0)) = \beta_0 + \zeta_{0i} +$$

$$\beta_b I(UV_i = Bounds) + \beta_t I(UV_i = Transparency) + \beta_a I(UV_i = Adjacent) +$$

$$\beta_L I(Rate_j = Low) + \beta_H I(Rate_j = High) +$$

$$\beta_{bL} I(UV_i = Bounds) * I(Rate_j = Low) + \beta_{bH} I(UV_i = Bounds) * I(Rate_j = High) +$$

$$\beta_{tL} I(UV_i = Transparency) * I(Rate_j = Low) + \beta_{tH} I(UV_i = Transparency) * I(Rate_j = High) +$$

$$\beta_{aL} I(UV_i = Adjacent) * I(Rate_j = Low) + \beta_{aH} I(UV_i = Adjacent) * I(Rate_j = High),$$

$$\zeta_{0i} \sim N(0, \sigma_\zeta)$$

We considered the estimated fixed effect coefficients and compared them with the confirmatory analysis. We then built
models with additional fixed effects (trial-level variables, participant-level covariates) and compared them to the baseline,
using model performance criteria. We only considered multilevel models with random intercepts (as a random "slope" on
condition would not be identifiable with the data). Thus, we simply added fixed effects stepwise to the baseline model to see if
they have better model fit, using multiple metrics.

A common model performance metric is the modified Bayesian Information Criterion (mBIC), which balances model good-
ness of fit with parsimony (Müller et al., 2013). Since we build classification (binary response) models, we also used classic
metrics for classification (defined in Table A1): correct classification rate (CCR); Brier score; and area under the Receiver
Operating Characteristic (ROC) curve (AUC). For all of these scores, we calculated the predicted probability of selecting the
location of higher uncertainty, $p_{ij}$, for a given participant and trial, based on the given model. We then binarized these predicted
probabilities at 0.5, obtaining binary predicted judgments $b_{ij}$ (is location of higher uncertainty predicted to be selected?). These
binary predicted judgments were then compared to the observed selections $o_{ij}$ with the goodness-of-fit metrics. This model se-


**Table A1.** Model performance metrics. $CCR$ is the correct classification rate, $AUC$ is the area under the ROC curve and use $n_t$, the number of trials and $n_p$, the number of participants. $mBIC$ is the modified Bayesian information criterion. For the definition of $mBIC$, $LL$ is the model log-likelihood, $f$ is the number of fixed effects and $r$ is the number of random effects estimated.

| Metric | Definition | Direction |
|---|---|---|
| $CCR(b_{ij}, o_{ij})$ | $CCR(b_{ij}, o_{ij}) = \frac{1}{n_t n_p} \sum_{i=1}^{n_p} \sum_{j=1}^{n_t} I(b_{ij} == o_{ij})$ | Higher is better |
| $AUC(p_{ij}, o_{ij})$ | Explained graphically in Fig. A4 | Higher is better |
| $Brier(p_{ij}, o_{ij})$ | $Brier(p_{ij}, o_{ij}) = \frac{1}{n_t n_p} \sum_{i=1}^{n_p} \sum_{j=1}^{n_t} (p_{ij} - o_{ij})^2$ | Lower is better |
| $mBIC(LL)$ | $mBIC = -2LL + \log(n_t n_p) \cdot (f + r)$ | Lower is better |

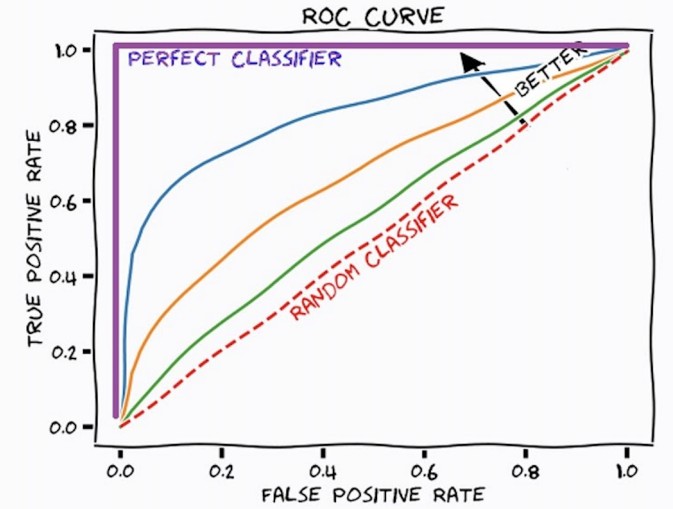

**Figure A4.** ROC curves as described in Serengil (2020). The area under the curve (AUC) for the best possible classifier should be as close to 1 as possible, as this would maximize the true positive rate and minimize the false positive rate.

lection analysis solely focused on identifying other variables beyond those in the baseline model that explain the experimental
data, based on in-sample goodness of fit. Significant effects in the best-fitting model could be considered as key determinants of task response and built into future experimental designs.

## Appendix B: Supplementary experimental results





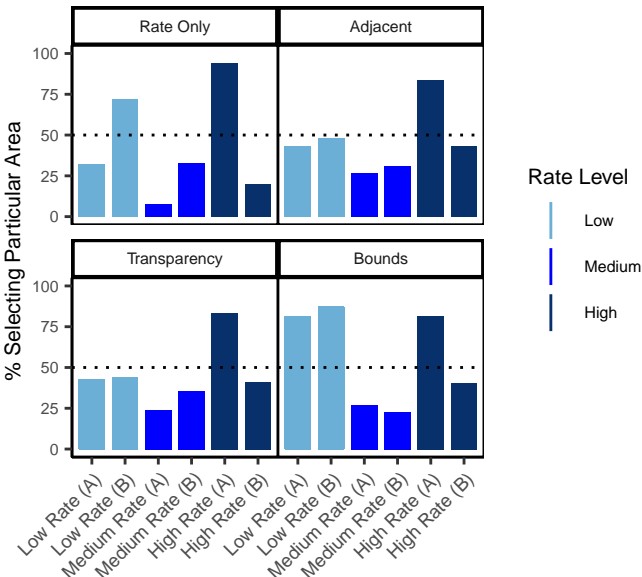

**Figure B1.** Percentage of participants selecting a particular location in the baseline trials, where both locations have the same rate and same low uncertainty. This is Location 1 in the baseline trials described in Table 2. Trials were repeated for forecast regions A and B.

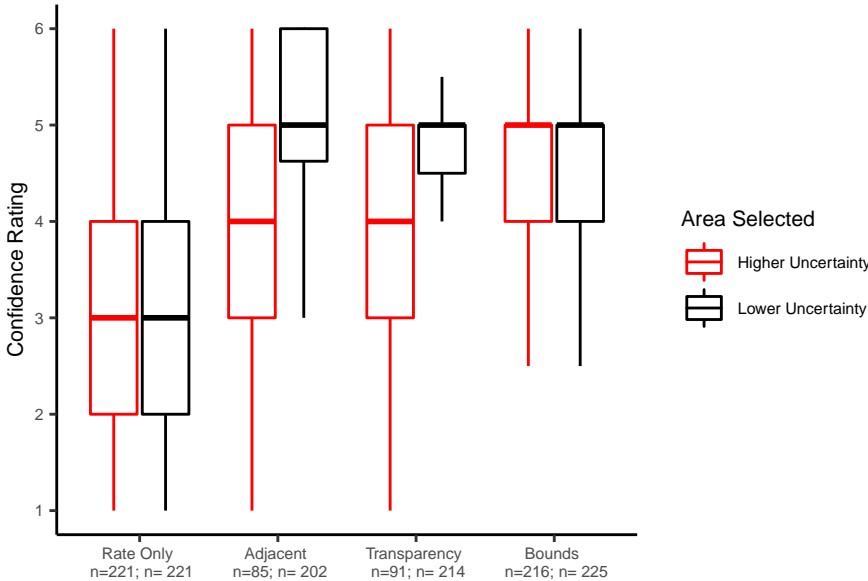

**Figure B2.** Boxplots for surprise judgment confidence ratings (median confidence by participant) by condition, binned between which location was selected. Sample sizes are given for each condition, the number of participants in that condition that selected the given location at least once.





**Table B1.** Most likely estimates and 95% confidence intervals (using Wald standard errors) for fixed effects in baseline model. The intercept is the logistic of the probability of selecting the higher-uncertainty location for the Rate Only condition for medium-rate trials (reference levels for visualization and rate level). Each fixed effect give show the change in the logistic of the probability corresponding to a change in visualization, rate level or their interaction (all else being held equal).

| Fixed Effect | Estimated Coefficient | 95% CI |
|---|---|---|
| Intercept | -1.07 | [-1.37, -0.76] |
| *Visualization* | | |
| Adjacent | -1.54 | [-2.04, -1.04] |
| Transparency | -1.90 | [-2.40, -1.40] |
| Bounds | 0.65 | [0.22, 1.07] |
| *Rate Level* | | |
| Low | 3.15 | [2.88, 3.41] |
| High | 0.64 | [0.43, 0.86] |
| *Visualization*Rate Level* | | |
| Adjacent*Low | -2.49 | [-2.92, -2.06] |
| Adjacent*High | -0.86 | [-1.27, -0.44] |
| Transparency*Low | -3.31 | [-3.75, -2.87] |
| Transparency*High | -1.43 | [-1.87, -0.99] |
| Bounds*Low | 0.01 | [-0.40, 0.42] |
| Bounds*High | -0.61 | [-0.92, -0.31] |

**Table B2.** Model performance for baseline model and other models with trial-level fixed effects. $CCR$ is the correct classification rate, $AUC$ is the area under the ROC curve and $mBIC$ is the modified Bayesian information criterion (see Table A1 for definitions).

| Model | Fixed Effects | $CCR$ | $AUC$ | $Brier$ | $mBIC$ |
|---|---|---|---|---|---|
| Model 1 | Visualization * Rate Level | 0.851 | 0.931 | 0.102 | 17072.6 |
| Model 2 | Visualization * Rate Level + Location Closer to Center | 0.858 | 0.941 | 0.094 | 16700.7 |
| Model 3 | Visualization * Rate Level + Location Closer to Center + Location Closer to High Rate Zone | 0.871 | 0.942 | 0.093 | 16670.9 |



**Table B3.** Model performance for baseline model and other models with participant-level fixed effects. $CCR$ is the correct classification rate, $AUC$ is the area under the ROC curve and $mBIC$ is the modified Bayesian information criterion (see Table A1 for definitions).

| Model | Fixed Effects | $CCR$ | $AUC$ | $Brier$ | $mBIC$ |
|---|---|---|---|---|---|
| Model 1 | Visualization * Rate Level | 0.851 | 0.931 | 0.102 | 17072.6 |
| Model 2 | Visualization * Rate Level + Age | 0.850 | 0.931 | 0.102 | 17078.6 |
| Model 3 | Visualization * Rate Level + log(numberEq + 0.01) | 0.851 | 0.931 | 0.102 | 17080.3 |
| Model 4 | Visualization * Rate Level + Education | 0.850 | 0.931 | 0.102 | 17112.6 |
| Model 5 | Visualization * Rate Level + Gender | 0.851 | 0.931 | 0.102 | 17089.9 |
| Model 6 | Visualization * Rate Level + State | 0.850 | 0.931 | 0.103 | 16948.4 |

*Author contributions.* We use the CASRAI Contributor Roles Taxonomy to categorize author contributions.

**Conceptualization**: MS, MM, PG, EAS, NF. **Resources** (developing uncertainty visualizations): MS, MM, PG, EAS, NF. **Methodology** (experiment and statistical analysis plan): MS, NF, MM, PG, EAS. **Investigation** (experimental data collection): NF, MS, MM. **Formal Analysis** (experimental data analysis): MS. **Writing - original draft**: MS. **Writing – review & editing**: MM, NF, PG, EAS. **Supervision**: MM, PG, EAS, NF.

*Competing interests.* The authors declare that they have no conflict of interest.

*Acknowledgements.* This research was supported by the Hans-Ertel-Centre for Weather Research, a research network of universities, research institutes and the German National Weather Service (DWD) funded by the BMVI (Federal Ministry of Transport and Digital Infrastructures). We gratefully acknowledge the five emergency management officials in Washington State, USA for informing our study; Andrew Michael, Joan Gomberg and Elenka Jarolimek for checks on visualizations and study design; Maik Messerschmidt and Antonio Amaddio, who custom-built our experimental platform; Alexandra Horsley and Jann Wäscher for assistance with piloting; and numerous colleagues at the Max Planck Institute for Human Development, Harding Center for Risk Literacy and University of Washington for fruitful discussions that developed this work.