# Peer review of "Effective uncertainty visualization for aftershock forecast maps"

_Natural Hazards and Earth System Sciences, 2021_

## Author Comment (AC1)

**Responses to Reviewer 1 for NHESS Article 2021-237: "Effective uncertainty visualization for aftershock forecast maps".**

Max Schneider, Michelle McDowell, Peter Guttorp, E. Ashley Steel, and Nadine Fleischhut

General comments

This paper addresses the highly relevant topic of support of judgments based on aftershock forecast maps. The authors describe an online experiment with over 880 participants, they compared map reading accuracy and judgment based on three different forecast map types including uncertainty. The choice of the three map types displaying forecast and uncertainty as adjacent maps, as coincident map with uncertainty as transparency, and as "bounds" (best case / worst case) maps makes sense from what we know from the literature. In advance, the authors conducted expert interviews for definition of the task which appropriate. Other positive aspects of the methodology are the pre-registration of the experiment, the determination of effect sizes from a pilot study, and the relatively high payment for the participants.

Results include differences in reading accuracy using the three maps, and participants' improved understanding of possible consequences in the high uncertainty case with the "bounds" representation, an implicit uncertainty visualization technique. The authors present a critical discussion of the results in light of the literature, with a focus on the most interesting judgment task. They give recommendations which map type should be used for which goal (accurate map reading or judgment under uncertainty). The findings are interesting and novel.

The formal quality of the manuscript is excellent, the language is clear and understandable.

Generally, I find this work to be of high quality, presenting new insights and practical implications for uncertainty visualization in the area of risk maps. I recommend its publication after addressing the issues I raise in the next section (minor revision).

*We thank the reviewer.*

Specific comments

The introduction is very good, the related work is well chosen and the separation into the three subsections (1.2, 1.3, and 1.4) makes sense to me. From subsection 1.1 and Figure 1 I understand that existing maps display the expected number of aftershocks per time or the probability of aftershocks. What I do not understand is the claim that the latter map "does not make uncertainty explicit" (l.88). I would say the probability of an aftershock at a location is an explicit measure for the uncertain event of an aftershock? I would like to ask the authors to clarify this point.

*We will make more clear around line 88 that a map showing estimated probabilities of a major aftershock (e.g., Fig. 1b) does not contain a measure of confidence in that estimate.*

With respect to the research questions in 1.5, I stumbled upon the term "lay users". This term does not seem sufficiently precise to me. I guess that the authors wanted to stress the fact that

map reading does not require domain knowledge. But why would you limit the observation to a group without this knowledge? In section 2 they even say that you deliberately chose participants from US states with earthquake events. So this obviously contradicts with the term "lay users". Please elaborate on this or think about a better term (maybe just "map readers" or "people"?).

*We will drop the distinction of (exclusively) lay users, as we agree that our experiment is designed to target communication goals that are relevant to users both with and without previous earthquake experience or from specific professional groups. We will explain the term "user" to mean any potential map reader and not exclusively one from a particular context (eg, professional group).*

What I see as a weakness in the experiment design is that task 3 depends on tasks 1 and 2 (reading accuracy). It should be ensured as a first experiment that all maps can be read with high accuracy. In a second experiment, only the maps fulfilling this criterion should be used for the judgment task. But maybe for this setup this is not a severe flaw. But the authors should add this to the limitations section.

*We will include a discussion of this point in the paragraph of the Discussions section that starts around line 652. Because we did not separately test the map-reading accuracy of visualizations before testing their effects on judgment, we could not fix potential issues in a visualization's design that may impede map-reading, and we can discuss this concern as a limitation.*

The finding that participants gain a better understanding of possible consequences in the high uncertainty case demonstrates the potential of intrinsic encodings, opposed to extrinsic ones. I suggest to add the terminology of intrinsic / extrinsic to the paper since it makes a difference here.

Technical corrections

I do not see the need for corrections of spelling errors or similar.

*We thank the reviewer.*

---

## Author Comment (AC2)

**Responses to Reviewer 2 for NHESS Article 2021-237: "Effective uncertainty visualization for aftershock forecast maps".**

Max Schneider, Michelle McDowell, Peter Guttorp, E. Ashley Steel, and Nadine Fleischhut

The paper "Effective uncertainty visualization for aftershock forecast maps" presents an empirical study of how different visualization styles affect interpretation of the uncertainty associated with aftershock forecast maps. Three uncertainty visualisations are tested through an online survey that was designed to address stakeholder communication needs for "sure bet" and "surprise" aftershock events. The task performance metrics reveal that presenting lower and upper bounds of uncertainty was more effective at communicating "surprise" events, where there is high uncertainty in the forecast.

Overall, the paper is very well written and addresses a timely and important topic in risk communication. The study design and data analysis are strong and the relevancy of the study is discussed in relation to the wider literature. It is excellent to see that the study design was informed by stakeholder needs, and the framing of exploring "sure bets" and "surprises" is novel and useful. I recommend this paper for further publication based on minor revisions that may help clarify some of the study design and its implications and limitations, as well as potentially improve digestibility of tricky uncertainty concepts.

*We thank the reviewer.*

Line 40-41: There are many previous experiments that consider user communication needs. I agree it is a strength of this study that interviews were carried out in order to explicitly inform the research design. However, that isn't immediately clear from this sentence. I would be more explicit and say that interviews were carried out with stakeholders to understand uncertainty visualisation needs so that the results can be used to inform evidence-based practice, or something along those lines.

*Our aim in lines 40-41 (in the abstract) was to highlight how our experiment differs from others (namely, that it uses tasks that are systematically designed to address broadly applicable and user-generated communication goals). We agree that previous experiments have considered user communication needs; we will remove this part from the sentence and frame a new sentence that explains that our experimental tasks are designed around communication goals that stem from user interviews.*

Line 86: Fig 1A needs a scale.

*There is a scale in Fig 1A but it was published without a title in the original source (Murru et al 2015); we will add a title to hopefully make this figure clearer.*

Line 95: the "skewed distribution" of aftershock rates is mentioned several times throughout the paper. I think it would be helpful to show an example plot of this distribution and it's associated uncertainty bounds. Uncertainty, especially when discussing probability distributions can be a bit of a tricky concept to digest – I think showing a graphic explanation of the "skewed" distribution,

it's upper and lower bounds, and visually highlighting areas where the "sure bet" and "surprise" events lie could help clarify the terms.

*We can make this suggested correction and add the described figure to the appendix.*

Line 138: Thompson et al. (2015) explore user interpretation of hazard curve graphics with upper and lower bounds alongside probability maps for ashfall forecast maps, though they don't explore the concept in depth or geospatially.

*We agree with the reviewer that Thompson et al. (2015) is an important study of the effects of uncertainty visualizations for volcanic hazards and uses upper and lower bounds designs, and we reference this study at several points in our paper (see lines 270 and 813), However, as the paper does not cover upper/lower bounds for maps, we do not reference it to support this particular point of the paper.*

Line 195: It may be worth mentioning that the higher uncertainty could also potentially result in fewer aftershock forecasts than forecasted?

*We can reformulate the sentence in question to explain that higher uncertainty means that both fewer or more aftershocks are possible than forecasted. Due to the skewness of the distribution of aftershocks, it is more likely that, between two locations of equal forecasted rates and different uncertainties, more aftershocks will occur at the location with higher uncertainty.*

Line 318, Fig 2: It would be helpful to break Fig 2 into part A and B, so that you can refer to the "rate only" map specifically, otherwise at first read it appears you are only showing 3 of the 4 visualisation conditions tested

*We can make this suggested correction.*

Line 325: The video tutorial explanation is a great approach to help ensure participants are fully informed before participating in the study

*We thank the reviewer.*

Line 452, Figure 6: I find this figure very difficult to interpret…Is there a way to simplify? The scale "% Judgement in favour of…" is not immediately clear…Is this between the two locations? Could consider using a graph design that enables you to show both locations for each trial (e.g., high and low rate; high and low uncertainty) for the reader to compare? It may be helpful throughout the figures to say "aftershock rate" instead of just "rate" for clarity.

*We thank the reviewer for the feedback and will split the figure into two separate panels. Then we can omit the legend entirely and allow each panel to have its own y-axis, saying, respectively "% judgment in favor of higher rate location", "% judgment in favor of higher uncertainty location". We can also reference the screenshots in the appendix in the figure caption; these are examples of the trials for this task.*

Line 545: Here the authors state that readers were less accurate reading uncertainty than other visualisations, but the abstract says at line 47 that all visualisations performed equally in communicating uncertainty…maybe double check consistency / language here, as I understand "sure bet" to still be communication of uncertainty

*We will make this correction to the text at line 545, making it clear that the Bounds visualization did worse in the map-reading task (map reading of uncertainty), not in the sure bets trials of the judgment task. However, the abstract says that all visualizations performed equally in communicating sure bets, not in the map-reading of uncertainty. We will tighten language to distinguish between these two things.*

Line 545 forward: Throughout the discussion the authors speculate on reasons for potential miscommunication, it would be worth noting in section 4.5 on limitations and future research that this study did not collect any open ended qualitative feedback, and that this may provide insight into some of these visual communication challenges.

*We did ask participants a single open-ended qualitative question at the end of the study, asking: "Remember the predictions you just made about which of two locations would have more aftershocks next week. Please try to formulate a rule on how you made your predictions. Another person should be able to follow that rule to make the same predictions as you."*

*We did not include analysis of this qualitative data in the manuscript as there was already a large body of results and discussions, but data for this question is in the study dataset uploaded to our OSF repository.*

Line 629-630: "Futhermore…" Can you clarify this sentence? Evenly binned in regards to what?

*This was a typo in the manuscript and will be corrected.*

Line 620: Can you discuss or note to what type of situation this might apply in practice?

*We can add an example from emergency management in the sentence ending on line 620.*

Line 672-673: The conclusion starts by stating that there are implications for non-technical users, then the following sentence outlines implications for technical users. Consider reframing/organising for clarity.

*We will replace the word "non-technical" with "a variety of".*

Line 680: The concluding paragraph talks about "intervals" and "intervals-based" design, but the results and discussion talk about "bounds", consistency would improve clarity of these statements

*We intentionally wanted to avoid using the terminology of the experiment (eg, Bounds) when making larger conclusions in the Conclusions section (eg, about interval-based uncertainty*

*visualization). We will however make it clearer at the beginning of the section that the Bounds visualization is an example of one that is interval-based.*

Lastly, the results about the bounds/interval map are very interesting but there is limited discussion of how these might inform design in practice (e.g., what are the possible implications of using two separate boundary maps labelled optimistic and pessimistic in a high stakes crisis communication environment? Could these be misinterpreted or separated?)

*We will add a sentence around line 667 discussing how the bounds maps might be useful in crisis management contexts.*

---

## Author Response (AR2)

**Responses to Editor for NHESS Article 2021-237: "Effective uncertainty visualization for aftershock forecast maps".**

Dear Drs. Kreibich, Malamud, Tarolli and Ulbrich,

Please find again enclosed our revised manuscript, "Effective uncertainty visualization for aftershock forecast maps" by Max Schneider, Michelle McDowell, Peter Guttorp, E. Ashley Steel, and Nadine Fleischhut, for publication in the journal *Natural Hazards and Earth Systems Sciences*.

The editor suggested to pay attention to two particular referee comments. Both of these were actually addressed in the revised manuscript; however, in our Author Response document, we unfortunately referred to the wrong (old) line numbers when mentioning where we made that revision. We do not believe we can actually expand more on these referee comments than we already have in the revised manuscript.

In the attached, please find our updated responses to the two referee comments that the editor spotlighted, with updated line numbers.

Thank you for considering our revisions.

Sincerely,

Max Schneider, Michelle McDowell, Peter Guttorp, E. Ashley Steel, and Nadine Fleischhut

Lastly, the results about the bounds/interval map are very interesting but there is limited discussion of how these might inform design in practice (e.g., what are the possible implications of using two separate boundary maps labelled optimistic and pessimistic in a high stakes crisis communication environment? Could these be misinterpreted or separated?)

*We will add a sentence around line 703-704 discussing how the bounds maps might be useful in crisis management contexts.*

What I see as a weakness in the experiment design is that task 3 depends on tasks 1 and 2 (reading accuracy). It should be ensured as a first experiment that all maps can be read with high accuracy. In a second experiment, only the maps fulfilling this criterion should be used for the judgment task. But maybe for this setup this is not a severe flaw. But the authors should add this to the limitations section.

*We will include a discussion of this point in the paragraph of the Discussions section that starts around line 674-676. Because we did not separately test the map-reading accuracy of visualizations before testing their effects on judgment, we could not fix potential issues in a visualization's design that may impede map-reading, and we can discuss this concern as a limitation.*